# WHICH ATTENTION HEADS MATTER FOR IN-CONTEXT LEARNING?

## ABSTRACT

Large language models (LLMs) exhibit impressive in-context learning (ICL) capability, enabling them to generate relevant responses from a handful of task demonstrations in the prompt. Prior studies have suggested two different explanations for the mechanisms behind ICL: induction heads that find and copy relevant tokens, and function vector (FV) heads whose activations compute a latent encoding of the ICL task. To better understand which of the two distinct mechanisms drives ICL, we study induction heads and FV heads in 12 language models.

Our study reveals that in all 12 models, few-shot ICL is driven primarily by FV heads: ablating FV heads decreases few-shot ICL accuracy significantly more than ablating induction heads, especially in larger models. We also find that FV and induction heads are connected: many FV heads start as induction heads during training before transitioning to the FV mechanism. This leads us to speculate that induction heads facilitate the learning of the more complex FV mechanism for ICL. Finally, the prevalence of FV and induction heads varies with architecture, which questions strong versions of the "universality" hypothesis: findings from interpretability research are not always generalizable across models[1].

## 1 INTRODUCTION

One of the most remarkable features of large language models (LLM) is their ability to perform in-context learning (ICL), where a handful of demonstrations provided during inference enables the model to perform various tasks. ICL is widely used and a crucial tool for steering pre-trained LLMs for specific tasks, and a growing body of work aims to understand the mechanisms behind ICL (Olsson et al., 2022; Akyürek et al., 2023; Von Oswald et al., 2023; Zhang et al., 2023).

To date, two key mechanisms have been primarily associated with ICL, substantiated by different lines of evidence. First, *induction circuits* (Elhage et al., 2021) were hypothesized to be the mechanism behind ICL in LLMs (Olsson et al., 2022; Singh et al., 2024; Crosbie & Shutova, 2024; Dong et al., 2022). Induction circuits perform ICL by looking back in the prompt for previous instances of the current token, then copying the subsequent token. More recently, Todd et al. (2024) and Hendel et al. (2023) propose the existence of *function vectors* (FV). FVs are a compact representation of a task extracted from a subset of attention heads in LLMs, and they can be added to a model's computation to recover ICL behavior without in-context demonstrations.

Are two seemingly distinct mechanisms both responsible for ICL in transformer LLMs? To better understand the role of induction circuits and FVs, we study the attention heads that implement induction and FVs (which we call *induction heads* and *FV heads*). We run a series of experiments on 12 decoder-only autoregressive transformer language models of parameter size between 70M and 7B (Table 2) and 48 natural language ICL tasks (listed in Appendix A.7).

We find that FV heads have the strongest causal effect on ICL performance on few-shot learning tasks. In Figure 1a, ablating FV heads leads to a significant drop in accuracy on ICL tasks, whereas ablating induction heads has a weak effect on ICL performance. This trend is consistent in all 12 models we studied, and FV heads are increasingly influential to ICL relative to induction heads as model scale increases (§4). This leads us to conclude that FV heads are mainly responsible for few-shot ICL, contrary to the prevailing belief that induction heads are a primary mechanism of few-shot ICL.

---

[1]Code and data will be released upon publication.

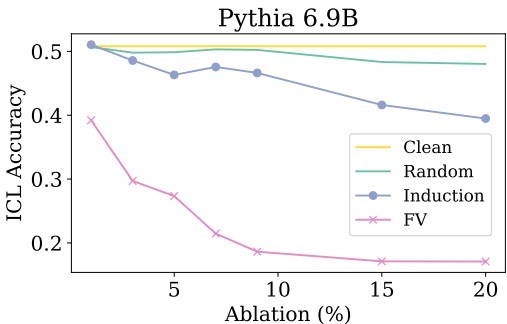 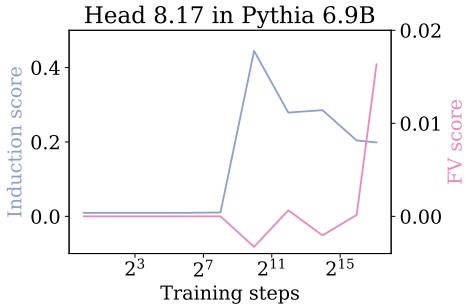

(a) ICL accuracy of Pythia 6.9B across different percentages of heads ablated. Ablating FV heads leads to the biggest drop in ICL accuracy.

(b) Induction score (blue) and FV score (pink) of an FV head during training. At around step $2^{11}$, this head has a high induction score. Then, its induction score decreases as its FV score increases.

Figure 1: The left figure shows that ablating function vector (FV) heads reduces model in-context learning (ICL) accuracy, whereas ablating induction heads does not affect ICL performance much more than ablating random heads, which suggests that FV heads matter most for ICL. The right figure shows an example of an FV head with high induction score earlier during training, which suggests that induction heads may be an early version of FV heads.

We identify two considerations that help explain the conflicting accounts in the literature (§2). Primarily, previous work connecting induction heads to ICL (Olsson et al., 2022) used a different metric for ICL, which we call *token-loss difference*. In our work, we measure ICL performance by computing accuracy on few-shot learning tasks, since this definition of ICL is more commonly adopted in the literature (Brown, 2020; Dong et al., 2022). We find that token-loss difference and few-shot accuracy behave differently, and ablating induction heads do strongly affect token-loss difference without similarly affecting few-shot accuracy.

Secondarily, studies on induction heads did not consider the correlation between induction heads and FV heads. Therefore, ablation studies that do not control for the correlation cannot compare the contributions of induction heads and FV heads to ICL well. Once we ablate only induction heads that are not also FV heads, the effect on ICL performance becomes much weaker. Previous studies were also often done on smaller models, since they required mechanistic analysis, but we find that FV heads become progressively more important for ICL relative to induction heads as models get larger.

We also further examine induction and FV heads in the context of each other, and find several relationships between them:

- The set of induction heads and FV heads are distinct, but there is some correlation between them (§3).

- FV heads appear deeper in models than induction heads (§3).

- FV heads appear later during training than induction heads (§5.1).

- There are many instances of induction heads that *transition to* FV heads during training, but the reverse does not occur (§5.1).

We summarize our findings in Table 1. In §6, we propose working conjectures to unify our findings. In one conjecture, we speculate that induction heads facilitate learning the more complex FV heads for ICL – the FV mechanism is more effective at performing ICL, and therefore eventually replaces the simpler induction mechanism. Our findings also question universality – the difference between the importance of FV heads and induction heads increases with model scale, where in our smallest model, the causal effect of FV and induction heads are similar. Our study underscores the variability of neural models and the importance of understanding the interplay between different mechanisms.

Table 1: Summary of findings in this work, where ✓ represents findings with evidence directly shown by our experiments and $\sim$ represents conjectures that our results suggest.

| Findings | Evidence | Section |
|---|---|---|
| Induction heads and FV heads are distinct | ✓ | 3 |
| Induction scores and FV scores are correlated | ✓ | 3 |
| Ablating FV heads hurts few-shot ICL accuracy more than ablating induction heads | ✓ | 4 |
| Some FV heads evolve from induction heads during training | ✓ | 5 |
| FV heads implement more complex or abstract computations than induction heads | $\sim$ | 5 |

## 2 BACKGROUND & RELATED WORK

We simultaneously investigate induction heads (Elhage et al., 2021; Olsson et al., 2022) and FV heads (Todd et al., 2024; Hendel et al., 2023) for a comparative analysis.

### 2.1 INDUCTION HEADS

Induction heads were first discovered by Elhage et al. (2021) and further investigated by Olsson et al. (2022) as the mechanism behind ICL. They are attention heads that attend to the token immediately after an earlier copy of the current token, and predict that the token attended to comes next.

Olsson et al. (2022) focus most of their study on small attention-only models with 1-3 layers, and finds co-occurrence between the emergence of induction during training and the emergence of ICL abilities in models, which they define as the loss of the 500th token minus the loss of the 50th token in the context. They also perform ablations and find that knocking out induction heads decreases ICL performance.

In our work, we analyze induction heads by computing the **induction score** of attention heads. We use the induction head detector in TransformerLens (Nanda & Bloom, 2022) to measure the induction score on a sequence of uniformly sampled random tokens $r_1 r_2 ... r_{50}$ that is repeated twice: $r = r_1 r_2 ... r_{50} r'_1 r'_2 ... r'_{50}$. The induction score for an attention head $a$ is given by:

$$S_I(a, r) = \sum_{i=1}^{50} a_{r'_i \to r_{i+1}}$$

where $a_{r'_i \to r_{i+1}}$ is the attention weight on token $r_{i+1}$ when processing token $r'_i$. For each attention head in each model, we take the mean induction score over 1000 samples of random sequences $r$, normalized by total attention mass to obtain a score between 0 and 1.

### 2.2 FV HEADS

Function vectors (FV) were concurrently discovered by Todd et al. (2024) and Hendel et al. (2023). FVs are a compact vector representation of ICL tasks extracted from certain attention heads, which can be added back to a language model's computations to trigger the execution of an ICL task. In our work, we call **function vector (FV) heads** the group of attention heads that transport function vectors.

We use the casual mediation analysis described in Todd et al. (2024) to identify FV heads that causally contribute to correctly performing few-shot ICL tasks. Given a set of natural language ICL tasks $\mathcal{T}$, where each ICL task $t \in \mathcal{T}$ defined by a dataset $P_t$ of in-context prompts $p_i^t \in P_t$ consisting of input-output pairs $(x_i, y_i)$, we first compute the mean activation of an attention head $a$ over prompts in $P_t$: $\bar{a}^t = \frac{1}{P_t} \sum_{p_i^t \in P_t} a(p_i^t)$.

Then, we run a transformer model $f$ on a corrupted ICL prompt $\tilde{p}_i^t \in \tilde{P}_t$, where each input $x_i$ is paired with a *random* output $\tilde{y}_i$. While running the model, we replace the activation of an attention head $a$ with the mean task-conditioned activation $\bar{a}^t$, and measure its **function vector score** (FV score) $S_{FV}$ as its causal indirect effect towards recovering the correct answer $y$ for the input $x$ given corrupted examples $(x_i, \tilde{y}_i)$:

$$S_{FV}(a|\tilde{p}_i^t) = f(\tilde{p}_i^t|a := \bar{a}^t)[y] - f(\tilde{p}_i^t)[y].$$

For each attention head, we take the mean FV score across 40 natural language ICL tasks from (Todd et al., 2024) (listed in Appendix A.7), where each task contains 100 prompts consisting of 10 input-output in-context examples and 1 input-output test pair.

## 2.3 RECONCILING DIVERGENT FINDINGS

While both mechanisms have been proposed by their respective works as the mechanism behind ICL, we find that if we analyze induction heads and FV heads side-by-side, FV heads seem to primarily contribute to ICL performance. We believe that the main reason for the divergence between our result and previous work lies in several intuitively related concepts in the literature that are assumed to be the same. On one hand, ICL is often used synonymously with few-shot learning from the prompt without parameter updates (Brown, 2020; Dong et al., 2022; Wei et al., 2023). We also adopt this conceptualization of ICL in this paper since it is the most standard in the literature, and for clarity, we will use "in-context learning" in this paper as equivalent to this definition. On the other hand, ICL performance is measured in Olsson et al. (2022) by computing the difference between the model loss of the 500th token in the context and the loss of the 50th token. This difference was previously called "ICL score" but we recommend adopting distinct terminology to avoid confusion, and for the purpose of our work, we will call this **token-loss difference**.

The primary discrepancy between our findings and Olsson et al. (2022) is that FV heads affect few-shot ICL accuracy strongly, but not token-loss difference, while induction heads affect token-loss difference strongly, but not few-shot ICL accuracy (§4). Secondly, we find that induction heads and FV heads are correlated (§3), and previous studies on few-shot ICL did not control for this correlation. In §4, when we naively ablated induction heads, the model significantly drops in ICL accuracy, which has been similarly shown in Crosbie & Shutova (2024) and Bansal et al. (2023). However, once we only ablate induction heads that are not also FV heads, their effect on ICL accuracy becomes close to random, whereas ablating FV heads that are not also induction heads still leads to a significant decrease in ICL accuracy. Therefore, previous studies that identified induction heads as important for ICL may be because they were observing signals from induction heads that are also FV heads. Finally, previous work that proposed induction heads as the mechanism driving ICL Olsson et al. (2022); Singh et al. (2024) focused on small model sizes to facilitate mechanistic analyses. However, we find that FV heads become increasingly important to ICL performance relative to induction heads as models increase in scale. In our smallest model with 70M parameters, induction and FV heads have similar causal effects to few-shot ICL, therefore it is important to study ICL mechanisms across a range of model scale.

Table 2: Models in this study. We use huggingface implementations (Wolf et al., 2020) for all models and load each model on an A4000 or A100 GPU. We report the number of parameters, number of layers $|L|$, total number of attention heads $|a|$, and the dimension of each head $\dim_a$ for each model.

| Model | Huggingface ID | Parameters | $|L|$ | $|a|$ | $\dim_a$ |
|---|---|---|---|---|---|
| Pythia (Biderman et al., 2023) | `EleutherAI/pythia-70m-deduped` | 70M | 6 | 48 | 64 |
| Pythia (Biderman et al., 2023) | `EleutherAI/pythia-160m-deduped` | 160M | 12 | 144 | 64 |
| Pythia (Biderman et al., 2023) | `EleutherAI/pythia-410m-deduped` | 410M | 24 | 384 | 64 |
| Pythia (Biderman et al., 2023) | `EleutherAI/pythia-1b-deduped` | 1B | 16 | 128 | 256 |
| Pythia (Biderman et al., 2023) | `EleutherAI/pythia-1.4b-deduped` | 1.4B | 24 | 384 | 128 |
| Pythia (Biderman et al., 2023) | `EleutherAI/pythia-2.8b-deduped` | 2.8B | 32 | 1024 | 80 |
| Pythia (Biderman et al., 2023) | `EleutherAI/pythia-6.9b-deduped` | 6.9B | 32 | 1024 | 128 |
| GPT-2 (Radford et al., 2019) | `openai-community/gpt2` | 117M | 12 | 144 | 64 |
| GPT-2 (Radford et al., 2019) | `openai-community/gpt2-medium` | 345M | 24 | 384 | 64 |
| GPT-2 (Radford et al., 2019) | `openai-community/gpt2-large` | 774M | 36 | 720 | 64 |
| GPT-2 (Radford et al., 2019) | `openai-community/gpt2-xl` | 1.6B | 48 | 1200 | 64 |
| Llama 2 (Touvron et al., 2023) | `meta-llama/Llama-2-7b-hf` | 7B | 32 | 1024 | 128 |

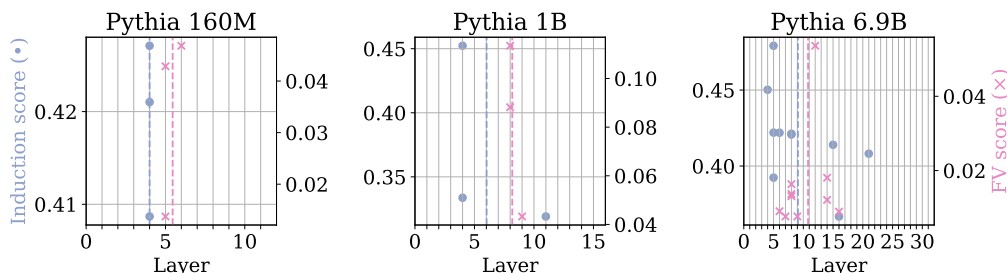

Figure 2: Location of induction heads (blue) and FV heads (pink) in model layers. The average layer of induction and FV heads are shown in blue and pink dotted lines respectively. Most induction heads appear in early-middle layers, FV heads appear at layers slightly deeper than induction heads.

## 3 INDUCTION HEADS AND FUNCTION VECTOR HEADS ARE DISTINCT BUT CORRELATED

Induction heads and FV heads implement two seemingly distinct mechanisms, and yet, they have both been attributed to ICL. To investigate whether one, or both, types of attention heads drive ICL, we start by studying the extent of the overlap between induction and FV heads.

### 3.1 HEAD LOCATIONS

First, we inspect where the top induction and FV heads are located in models to understand whether they appear in similar layers of the model. In Figure 2, we plot the layers where the top 2%[2] induction heads and FV heads appear in 3 Pythia models, with the induction score of induction heads on the left y-axis and FV score of FV heads on the right y-axis. We plot the head locations for all 12 models in Appendix A.9.

In general, induction heads appear in early-middle layers and FV heads appear in slightly deeper layers than induction heads. This suggests that induction and FV heads may not fully overlap, and that FV heads may implement more complex or abstract computations than induction heads.

### 3.2 OVERLAP BETWEEN INDUCTION AND FV HEADS

Induction heads and FV heads mostly appear in distinct model layers, but there are a few layers that contain both induction and FV heads where there could be potential overlap between the two types of heads. We therefore investigate the extent of overlap between the two types of heads in Figure 3.

First, we plot the overlap between the top 2% induction heads and FV heads: $100 \times \frac{|IH \cap FV|}{|IH|}$ where $IH$ and $FV$ are the set of top induction heads and FV heads respectively (Figure 3 left). 7 out of 12 models have no overlap between induction and FV heads, and others have some overlap between 5-15% of induction / FV heads. This leads us to conclude that **induction heads and FV heads are mostly distinct**.

To further investigate the connection between induction and FV heads beyond the top 2% heads, we also compute the percentile of the induction score of the top 2% FV heads (Figure 3 center) and the percentile of the FV score of the top 2% induction heads (Figure 3 right). In most models, FV heads are at around the 90-95th percentile of induction scores, and vice versa. Therefore, although there is little overlap between induction and FV heads, **induction and FV scores are correlated**: FV heads have high induction scores compared to other attention heads, induction heads have high FV scores. [3]

---

[2]In certain cases, we need to differentiate between meaningful induction / FV heads and the long tail of attention heads that perform neither induction nor FV mechanism. In this work, we choose the top 2% induction and FV heads as the representative set of induction and FV heads, following Todd et al. (2024).

[3]In our main analysis, we do not rely on the correlation between the distribution of induction scores and FV scores across the full set of attention heads because there is a long tail of attention heads with low scores on both induction and FV. For completeness, we plot the induction and FV scores of all heads in Appendix A.1.

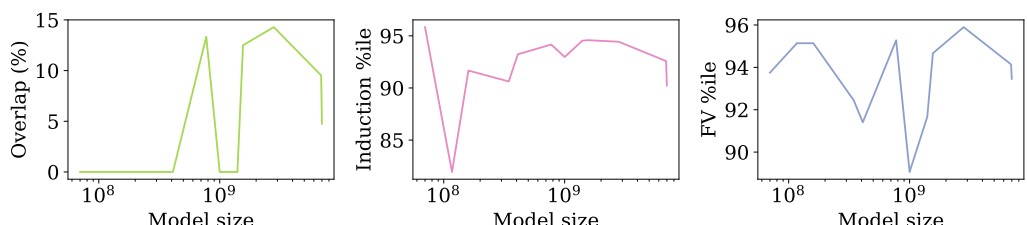

Figure 3: Percentage of head overlap between induction and FV heads (left). Percentile of induction score of FV heads (center). Percentile of FV score of induction heads (right). There is little overlap between induction and FV heads, but FV heads have relatively high induction scores and vice versa.

## 4 FUNCTION VECTOR HEADS DRIVE IN-CONTEXT LEARNING

If induction heads and FV heads are distinct from one another, which type of attention heads matter most for ICL? To investigate this, we measure the causal importance of different heads for ICL by evaluating ICL performance after ablating induction and FV heads. We also control for the correlation between induction and FV heads by ablating induction heads while preserving FV heads, and vice versa. We measure ICL performance by evaluating model accuracy on few-shot ICL tasks. For completeness, we also study the effects of ablation on token-loss difference.

### 4.1 METHOD

**Ablation.** To analyze the causal effect of specific attention heads on ICL performance, we study how much ICL performance drops when we "erase" certain heads. To do so, we perform mean ablation on groups of heads by replacing the output vector of each head we ablate by the mean of this head's outputs over all examples in our dataset of few-shot ICL tasks described in later sections. We choose to perform mean ablation instead of zero ablation (setting the output to 0) to avoid the out-of-distribution problem Hase et al. (2021); Wang et al. (2023); Zhang & Nanda (2024).

Since FV and induction scores are correlated, there may be overlap between the heads we ablate for induction and FV mechanisms, especially in larger ablation quantities. This may obfuscate the comparison of contributions between the two types of heads. We therefore also perform ablation experiments with "exclusion": when ablating $n$ FV heads, we take the top $n$ heads by FV score that do not appear in the top $2\%$ heads by induction score, and similarly when ablating $n$ induction heads, we take the top $n$ heads by induction score that do not appear in the top $2\%$ heads by FV score.

**Few-shot ICL accuracy.** We evaluate ICL performance by measuring model accuracy on a series of few-shot ICL tasks. Each ICL task is defined by a set of input-output pairs $(x_i, y_i)$. The model is prompted with 10 input-output exemplar pairs that demonstrate this task, and one query input $x_q$ that corresponds to a target output $y_q$ that is not part of the model's prompt. We compute the model's accuracy in predicting the correct output $y_q$. We describe the full set of ICL tasks we study in Appendix A.7.

To avoid leakage between ICL tasks used to identify FV heads and those used to evaluate FV head ablations, we randomly split the 40 ICL tasks from Todd et al. (2024) into 29 tasks used to measure FV scores of heads, and 11 tasks to evaluate ICL performance. We also add 8 new tasks for ICL evaluation: 4 tasks are variations of tasks in Todd et al. (2024), and 4 are binding tasks from Feng & Steinhardt (2024). In total, we evaluate ICL accuracy on 19 natural language tasks, with 100 prompts per task.

**Token-loss difference.** To compare with previous work, we also study the effect of ablations on token-loss difference used in Olsson et al. (2022). We measure token-loss difference by taking the loss of the 50th token in the context minus the loss of the 500th token in the context[4], averaged over 10,000 examples from the Pile dataset (Gao et al., 2020).

---

[4]We flip the difference used in Olsson et al. (2022) so that a higher score indicates better ICL performance

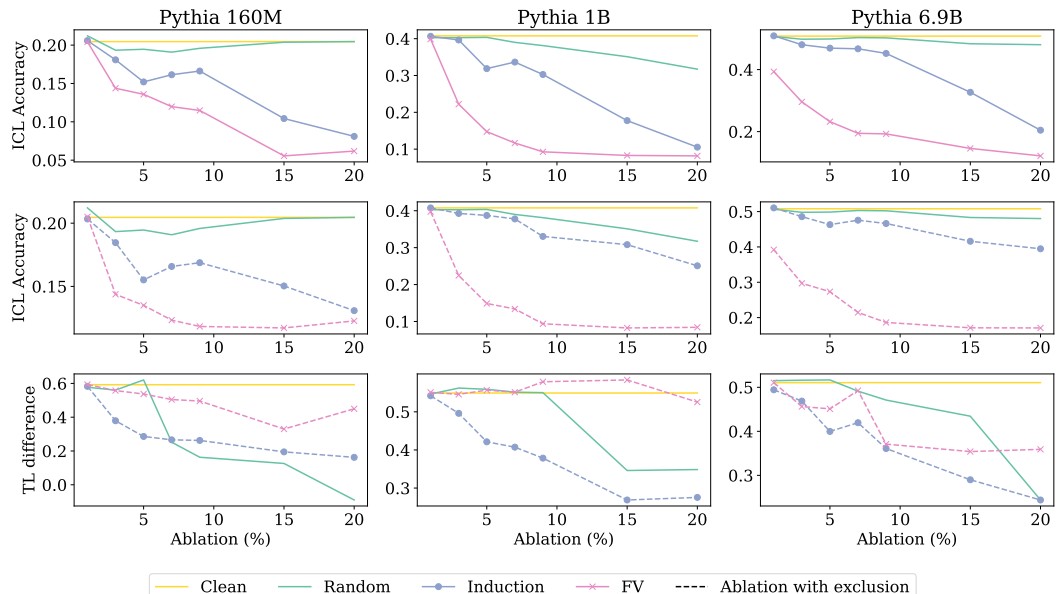

Figure 4: Top: ICL accuracy after ablating induction and FV heads. Center: ICL accuracy after ablating non-FV induction and non-induction FV heads. Bottom: Token-loss difference after ablating non-FV induction and non-induction FV heads. Ablating FV heads lead to a bigger drop in ICL accuracy, especially in larger models. Ablating induction heads with low FV scores does not significantly affect ICL accuracy. ICL accuracy and token-loss difference behave differently.

## 4.2 RESULTS

For each model, we ablate the top 1-20% of attention heads based on induction or FV score. We also compute model performance with no ablation and with ablations of randomly sampled heads as a baseline. In Figure 4, we plot how model ICL accuracy and token-loss difference evolve with different quantities of ablations for a sample of 3 models, where ICL accuracy is averaged over the 19 tasks used for evaluation. We also plot ablations for all models, and ICL accuracy broken down by task, in Appendix A.8.

In the top row of Figure 4, we ablate induction and FV heads without excluding one from the other. Here, we find that in general, ablating FV heads leads to a bigger drop in ICL performance than induction heads, and the gap between the effect of FV heads and induction heads is larger in bigger models. We also find that ablating induction heads leads to worse ICL accuracy than ablating random heads, and the effect of ablating induction heads converges to the effect of ablating FV heads when we increase the number of heads ablated.

However, the convergence noted above may be due to an increasing overlap in the set of heads ablated in the induction head and FV head ablations (Appendix A.10). We control for this overlap using ablations with exclusions. The center row of Figure 4 shows that after ablating induction heads while preserving FV heads, the effect of ablating induction heads on ICL is weaker. In fact, in most models with parameter count over 1B, the effect of ablating induction heads is close to ablating random heads. On the other hand, ablating FV heads while preserving induction heads still causes a large drop in ICL performance. We also recover a similar trend as before across model scale: the larger the model, the larger the gap between the effect of FV heads and induction heads. This suggests that the contributions of induction heads to ICL in the top row of Figure 4 mostly come from heads that are both induction and FV heads, and that **FV heads matter the most for ICL**: as long as the model preserves its top 2% FV heads, it is able to perform ICL with reasonable accuracy even if we ablate induction heads.

In the bottom row of Figure 4, we report the effect of ablation on token-loss difference after excluding FV heads from induction head ablations and vice versa. Here, we find that in smaller models (less than 160M parameters), ablating induction or FV heads does not influence token-loss difference more

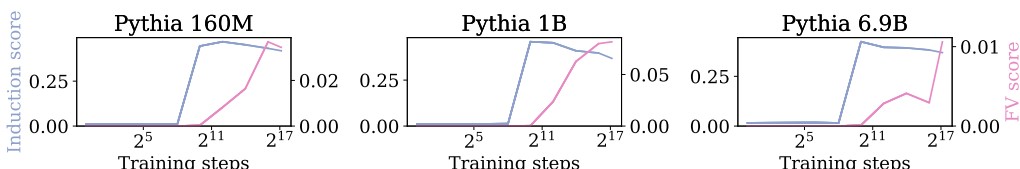

Figure 5: Evolution of induction score and FV score averaged over top 2% heads across training. Induction score rises sharply, then plateaus. FV score rises slightly later than induction scores, and steadily increases throughout training.

than random. For models with more than than 345M parameters, ablating induction heads lead to a larger drop in token-loss difference than ablating FV heads, but the gap between the effect of ablating induction and FV heads decreases with model scale. This experiment shows that ICL accuracy and token-loss difference measure two very different things, which helps explain the discrepancy in the literature.

## 5    FV HEADS EVOLVE FROM INDUCTION HEADS

Our previous findings on the correlation between induction and FV scores hint at some interplay between induction and FV heads. To better understand how the two types of heads may be related, we examine how induction and FV heads evolve during training in 8 intermediate training checkpoints of 7 Pythia models.

### 5.1    INDUCTION AND FV STRENGTH DURING TRAINING

To measure the general strength of induction and FV mechanisms over the course of training, we plot the mean induction and FV scores of the top 2% induction and FV heads at each model checkpoint (Figure 5). We include plots for all Pythia models in Appendix A.11.

In all Pythia models, induction heads appear at around step 1000 out of 143000 during training, whereas FV heads appear later at around step 16000 out of 143000. Moreover, induction score rises sharply at step 1000 and then plateaus or slightly decreases, whereas FV score gradually increases from step 14000 until the end of training. This suggests that **induction heads are easy for models to learn and FV heads are harder to learn.**

### 5.2    EVOLUTION OF INDIVIDUAL HEADS DURING TRAINING

We further investigate the evolution of induction and FV scores of individual attention heads. In Figure 6, we plot the induction scores (top row) and FV scores (bottom row) of the top 2% induction and FV heads across training steps. In each model, we find that certain FV heads have high induction score earlier in training, at around the same time as when induction heads form, and often matching the induction score of induction heads. These FV heads then observe a decay in induction score while increasing in FV score later in training. However, the reverse is not true: all induction heads have low FV scores throughout training. This suggests that **many FV heads evolve from induction heads during training.**

## 6    INTERPRETATION AND DISCUSSION

To summarize our key findings, we verified that induction heads and FV heads are distinct, but correlated (§3). FV heads appear in slightly deeper layers than induction heads, and emerge later during training (§3,5.1). To the extent that these two types of heads are different, FV heads play a large role in ICL, especially in bigger models, whereas ablating induction heads does not decrease ICL performance to the same extent (§4). We also find several instances of induction heads that transition to FV heads during training, whereas the reverse does not occur (§5.1). We propose two

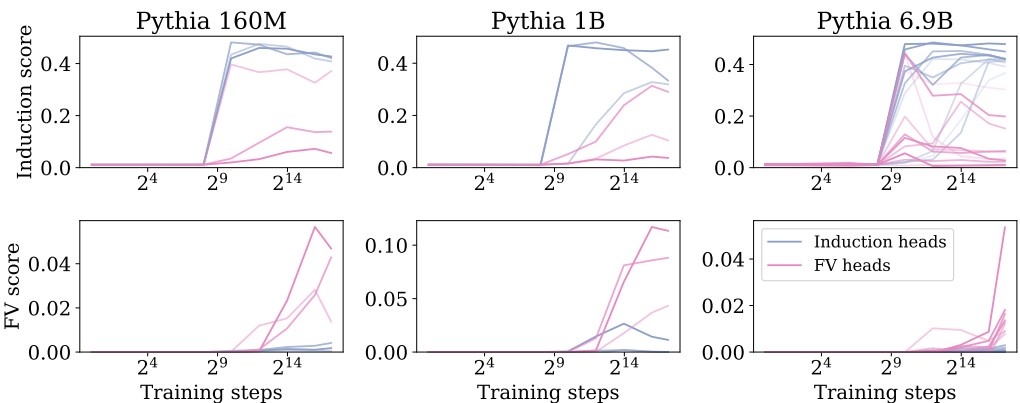

Figure 6: Evolution of induction scores (top) and FV scores (bottom) of individual induction and FV heads across training. Certain FV heads have a high induction score earlier in training; the reverse is not true for induction heads.

working conjectures to explain these findings more broadly, and consider arguments for and against them.

Our first conjecture (C1) is that **induction heads are an early version of FV heads**. We believe induction heads help the model learn the more complex FV heads. Once the model obtains FV heads, the FV mechanism is more accurate for performing ICL and therefore it eventually replaces the simpler, less effective induction mechanism.

The most compelling argument for C1 is the existence of FV heads that are formerly induction heads during training. The decrease in induction score late in training suggests that induction is not needed by the head in its final form. This is reinforced by the fact that the reverse never happens (FV heads never become induction heads), and that induction heads with low FV scores contribute little to ICL under ablations. To further verify this, future work could explore how removing induction heads during training could impact the development of FV heads. However, C1 does not fully explain the existence of FV heads with low induction score throughout training.

C1 also correctly predicts that FV heads are more complex and harder to learn than induction heads (they appear later in training and deeper in model architectures). This also explains why attention heads with higher parameter count are needed to learn strong FV heads, and ablating FV heads affect larger models more than smaller models relative to ablating induction heads.

An alternative conjecture (C2) is that **FV heads are a combination of induction and another mechanism**. Under this conjecture, the induction heads that "become" FV heads are polysemantic heads that implement both induction and FVs, and possibly other mechanisms. The induction score of these polysemantic heads drop because the attention patterns become split between the different mechanisms the attention head carries out. Under C2, induction and FV heads are correlated because they share an underlying mechanism that the model learns to re-use for both tasks during training. However, C2 would predict that ablating monosemantic FV heads would not hurt ICL performance, whereas we observe that ablating monosemantic FV heads while preserving polysemantic induction-FV heads lowers ICL accuracy.

## 7  CONCLUSION

Contrary to the prevailing consensus that few-shot ICL is largely driven by induction heads, we find that this assumption does not hold in most of the models we study. Instead, we find that FV heads have a more important causal contribution to few-shot ICL. We believe the main reason for this misconception is due to conflating few-shot prompting and token-loss difference when we discuss ICL, as well as not accounting for the overlap between induction and FV heads.

Remarkably, although induction and FV mechanisms appear to implement two distinct processes, we also observe interesting interplay between the two types of heads: induction and FV scores are correlated, and many FV heads are 'former' induction heads that have high induction scores earlier in training. In §6, we present arguments for and against early conjectures to explain this phenomenon. We believe that one possible explanation is induction heads act as a precursor to FV heads: induction, being simpler to learn and implement, initially facilitates ICL, from which the FV mechanism emerges to achieve a more accurate implementation of ICL.

Our study also illustrates an important takeaway for interpretability research more generally. In several experiments, we find that certain models with different architecture, model family, or parameter size behave qualitatively differently than others. For example, induction heads significantly contribute to ICL in small Pythia models but not in larger Pythia models (§4). The invalidity of the strong version of the universality hypothesis in interpretability could also explain why induction heads were previously commonly attributed to ICL: early studies focused on small models to facilitate mechanistic analysis, which may give an incomplete explanation of ICL mechanisms.

Our findings open up several avenues for future investigation. If C1 is true, why do FV heads need induction as a precursor? What are remaining induction heads used for? Is there a third mechanism that better explains ICL? If model mechanisms are not universal across size or architecture, what is the best approach to generalizing findings from interpretability research?

## REPRODUCIBILITY STATEMENT

We recognize the importance of reproducibility and have made the following efforts to ensure reproducibility of our findings. First, we detail the huggingface IDs of all the models we use, as long as the machines we conducted all experiments on, in Table 2. Second, we define how we computed induction and FV scores of attention heads, including the public packages we used, in §2. Third, we also describe the datasets we used in §4. Lastly, we will release all code and data needed to reproduce all findings and figures in our work upon publication.

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

## A  APPENDIX

### A.1  INDUCTION SCORES VS. FV SCORES

In Figure 11, we plot the induction score and FV score of each attention head.

### A.2  ABLATIONS

In Figure 7, we plot model accuracy averaged over ICL tasks across different quantities of heads ablated in each head type. In Figure 8, we plot the token-loss difference of models across different quantities of heads ablated.

### A.3  RANDOM AND ZERO ABLATIONS

In Figure 9, we plot model accuracy averaged over ICL tasks across different quantities of heads ablated with random ablation or zero ablation. For random ablations, we replace the head's output vector with the output vector of a randomly sampled different head. For zero ablations. we replace the head's output vector with a zero vector.

### A.4  ABLATING RANDOM HEADS AT SPECIFIC LAYERS

In Figure 10, we ablate heads randomly sampled from specific layers of the model. Let $L$ be the number of heads in each layer, $A$ be the number of heads we're ablating, and $\ell$ be the layer we're targeting. Then, if $A < L$, we sample $A$ heads from layer $\ell$. If $A \geq L$, we ablate all $L$ heads in layer $\ell$ and we sample $A - L$ heads from other layers to ablate.

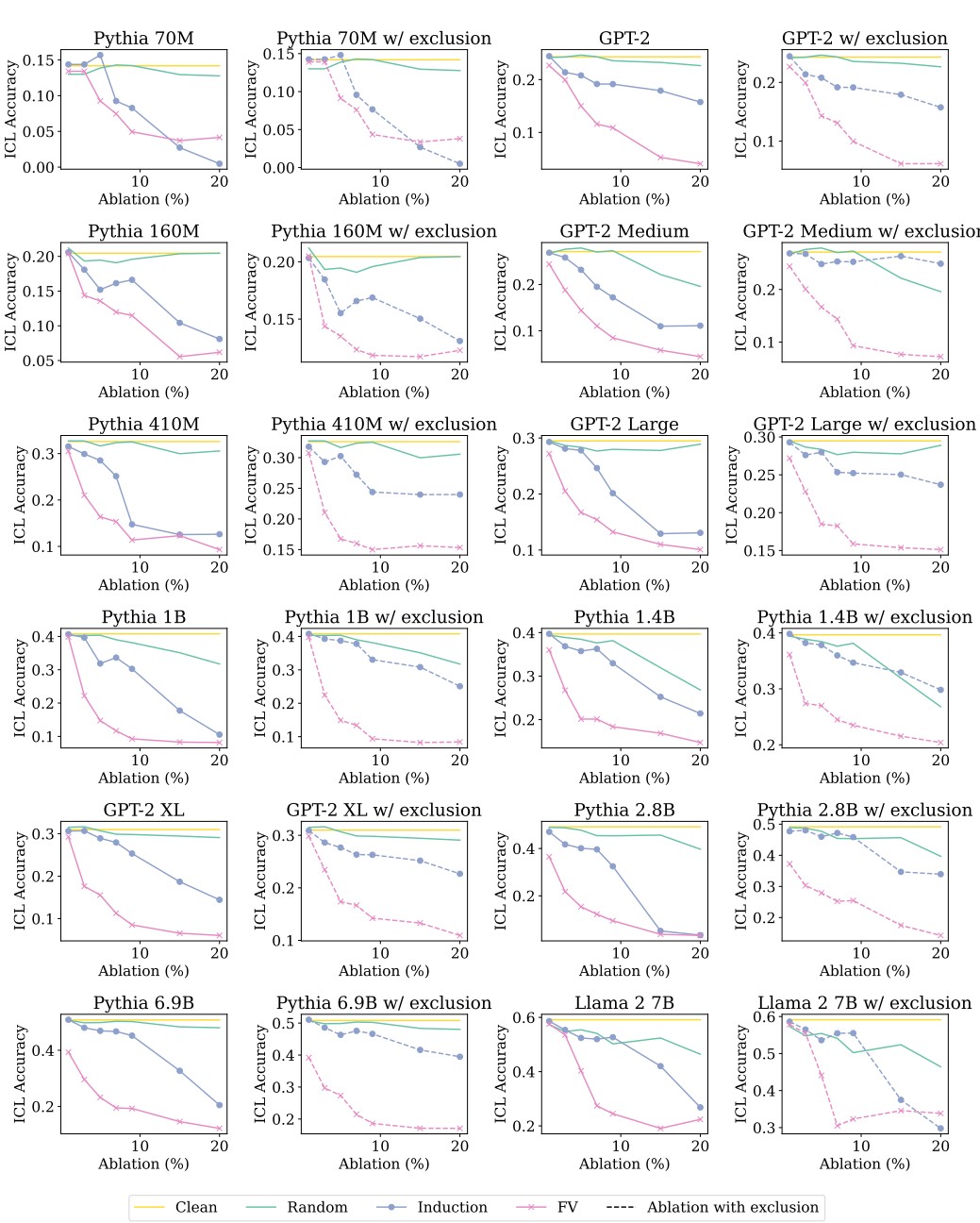

Figure 7: ICL accuracy after ablating induction and FV heads.

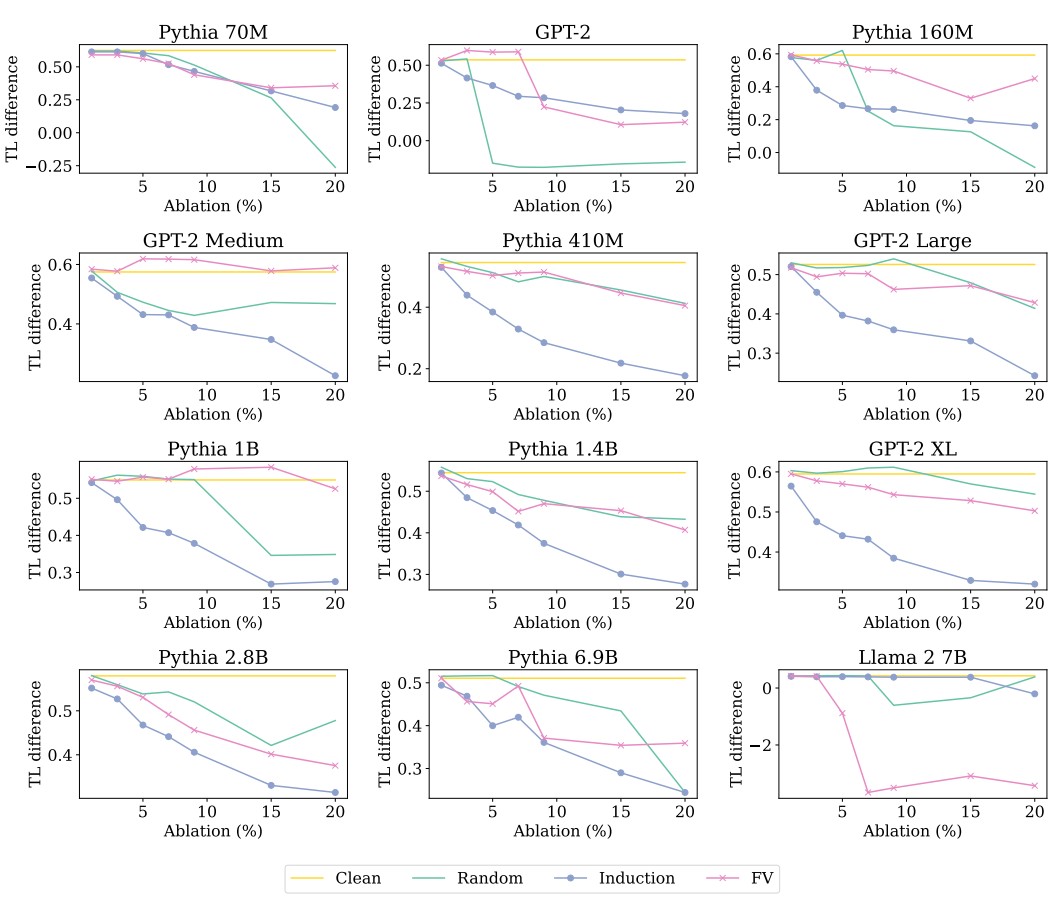

Figure 8: Token-loss difference after ablating induction heads with low FV scores and FV heads with low induction scores.

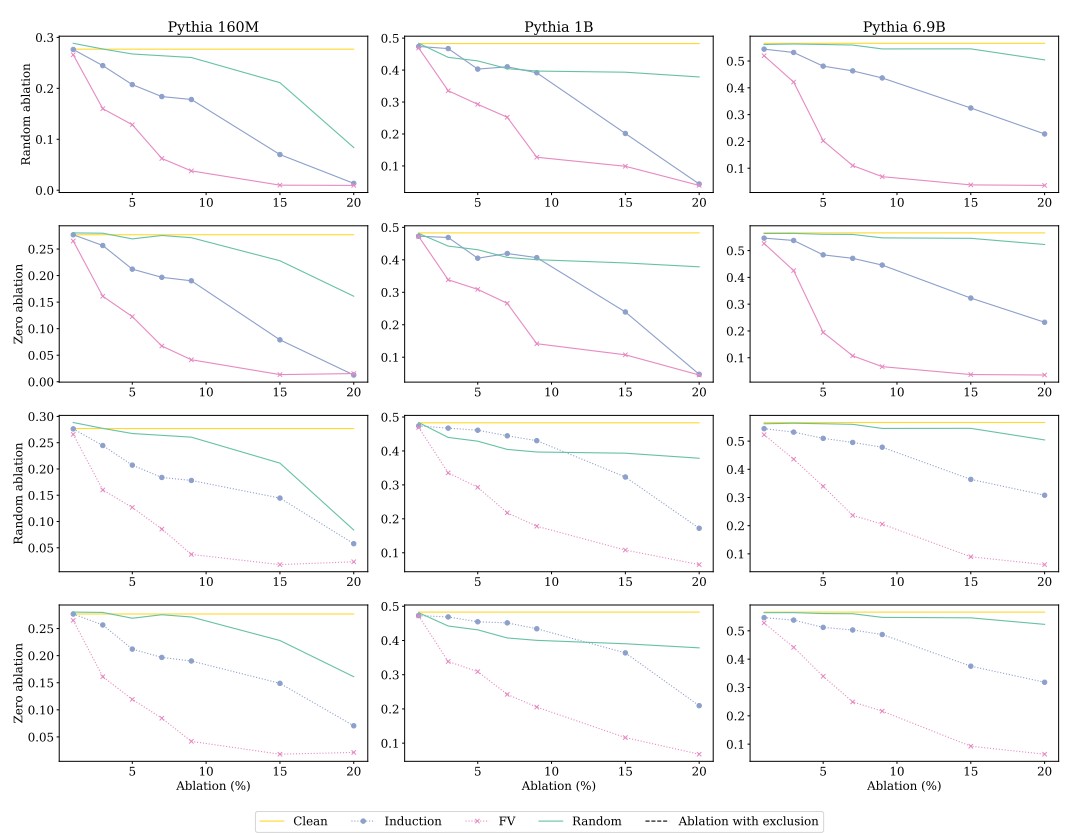

Figure 9: ICL accuracy after ablating induction heads and FV heads with random or mean ablation.

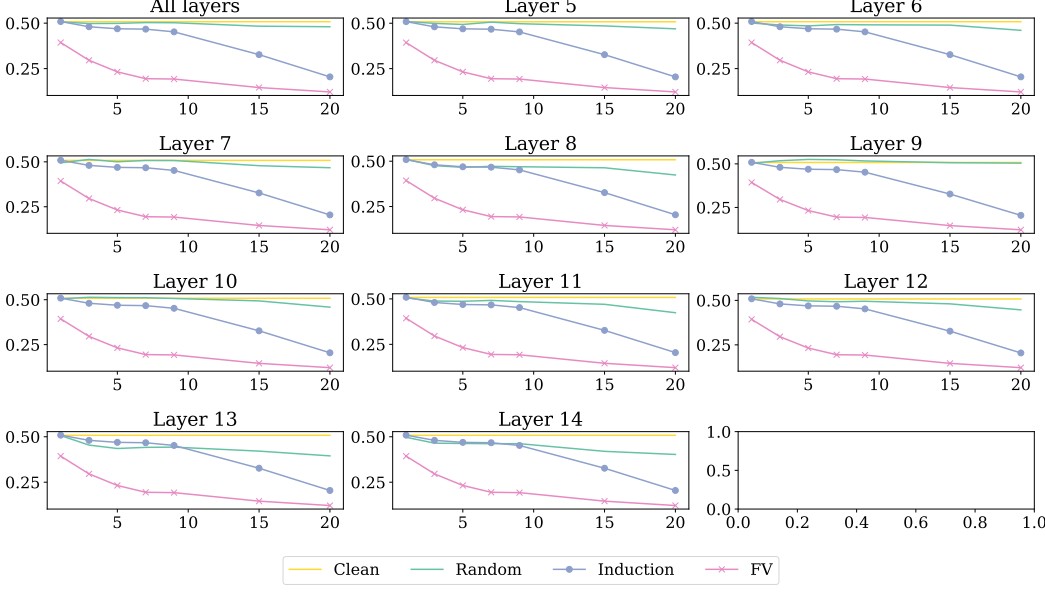

Figure 10: ICL accuracy after ablating randomly sampled heads from specific layers. The clean ICL accuracy, induction ablations and FV ablations are also plotted for comparison but only the random ablations (green curve) are affected by the choice of target layer.

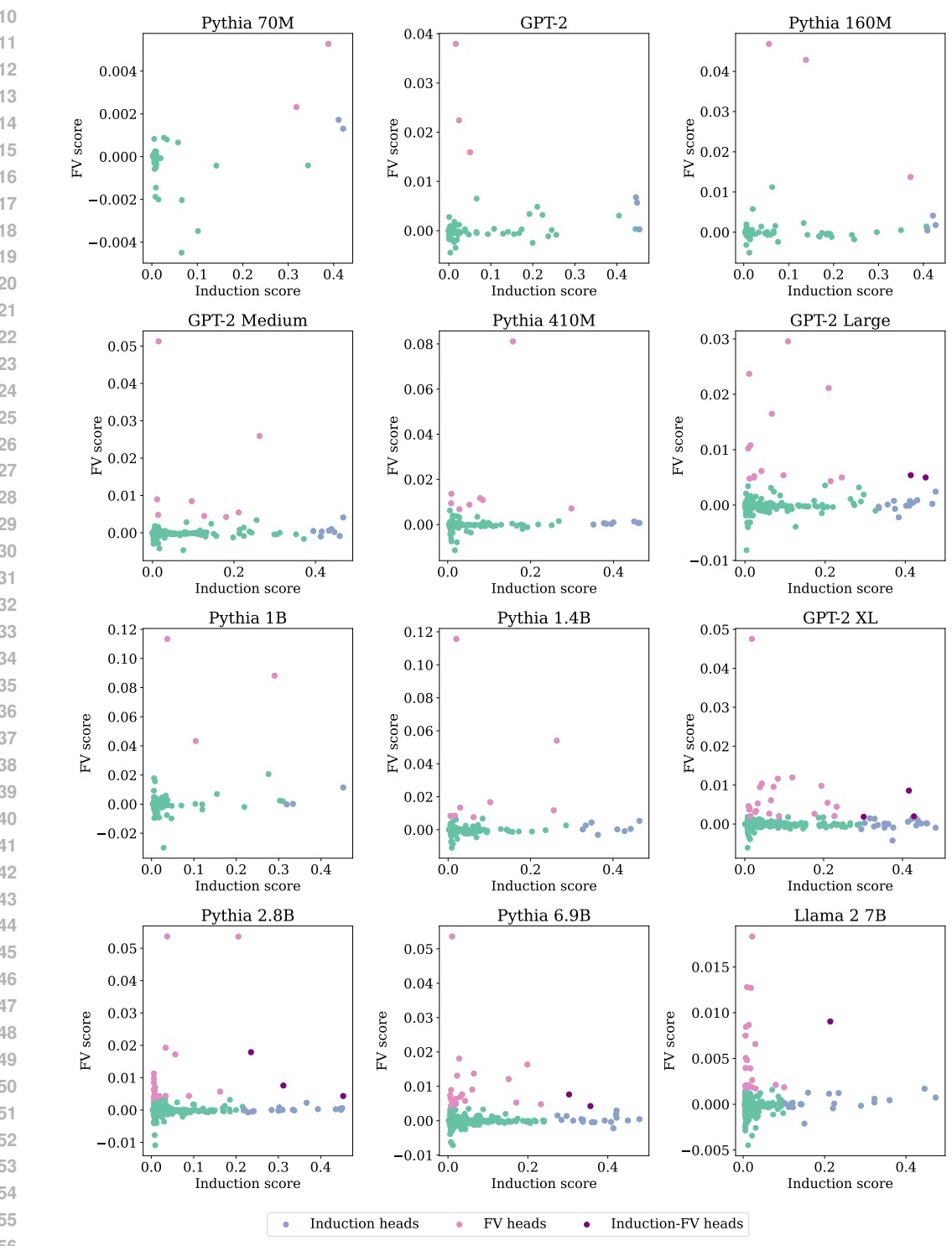

Figure 11: Induction and FV scores of attention heads.

## A.5 INDUCTION AND FUNCTION VECTOR SCORES ACROSS MODELS

Our ablation studies reveal a consistent trend where FV heads are increasingly important relative to induction heads for ICL performance as model scale increases. To further explore this trend, we

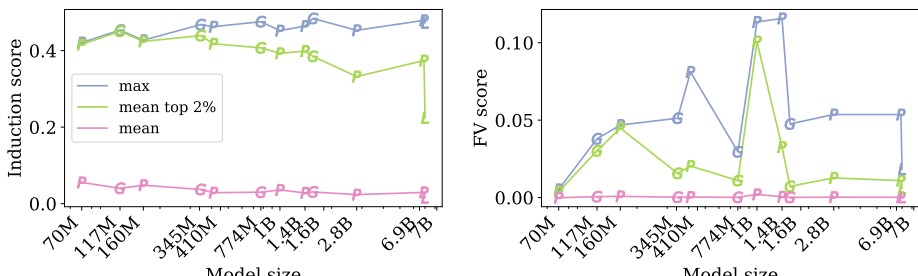

Figure 12: Induction score (left) and FV score (right) of attention heads across model size. We plot the maximum score of all heads, mean of the top 2% scores, and mean score of all heads. Overall, induction scores are similar across models. Pythia 70M and Llama 2 have relatively low FV scores, Pythia 1B and 1.4B have relatively high FV scores.

examine how induction scores and FV scores vary with model scale, and whether these scores follow similar trends to our ablation experiments.

In Figure 12, we plot the maximum and mean induction and FV scores across all heads, and mean scores of top 2% heads, for each model. The left plot in Figure 12 shows that induction scores are relatively similar across model size, with a small increase in maximum induction score and a decrease in the top 2% mean induction score with model scale.

In the right plot of Figure 12, there is no clear trend between FV score and model scale, however, Pythia 1B and 1.4B models have markedly higher maximum FV scores. One possible explanation is that models with high head dimensionality relative to total parameter count have stronger FV heads: Pythia 1B and 1.4B have head dimensionality of 256 and 128 respectively (Table 2) whereas other models with similar parameter count have only 64-80 attention head dimensions.

We also find very low FV scores in Pythia 70M and Llama 2 models. FV scores may be low in Pythia 70M because it is too small in parameter size for FV heads to emerge. Low scores in Llama 2 compared to other models may be due to differences in architecture, and additional experiments can help confirm this. Overall, we do not recover the same trend in induction/FV scores as the trend in our ablation studies.

For reference, we also provide box plots of the full distribution of induction and FV scores in Figure 13.

### A.6    EVALUATING FUNCTION VECTORS ON TASK EXECUTION

To further inspect the prevalence of the FV mechanism in different models, we evaluate the efficacy of FVs for ICL task execution. A successful FV triggers the model to execute the particular task the FV encodes, even when the model sees no useful in-context demonstrations of the task. First, to extract FVs, for each model we gather the top 2% attention heads with highest FV scores as the set $\mathcal{A}$. Then, for each ICL task $t \in \mathcal{T}$, we sum the average outputs of heads in $\mathcal{A}$ over prompts from $t$ and obtain the FV for the task $t$: $FV_t = \sum_{a \in \mathcal{A}} \bar{a}^t$.

In Figure 14, we report model accuracy averaged over 40 ICL tasks where the model performs inference on uncorrupted prompts (clean), prompts with shuffled labels (shuffled), shuffled prompts with $FV_t$ added to hidden states at layer $|L|/3$, and shuffled prompts with FV extracted from random heads added to hidden states at layer $|L|/3$. We take 1000 examples per task that are previously unseen during FV score computation.

In most models, adding the FV recovers model performance on uncorrupted prompts, with the exception of Pythia 2.8B. One possible explanation for this is again due to head dimensionality: Pythia 2.8B has head dimension 80, which is significantly smaller than other models with similar parameter size that have head dimensions of 128. Together with our experiments in §A.5, results provide preliminary evidence that **high head dimensionality relative to model size is a predictor of FV strength (H6)**.

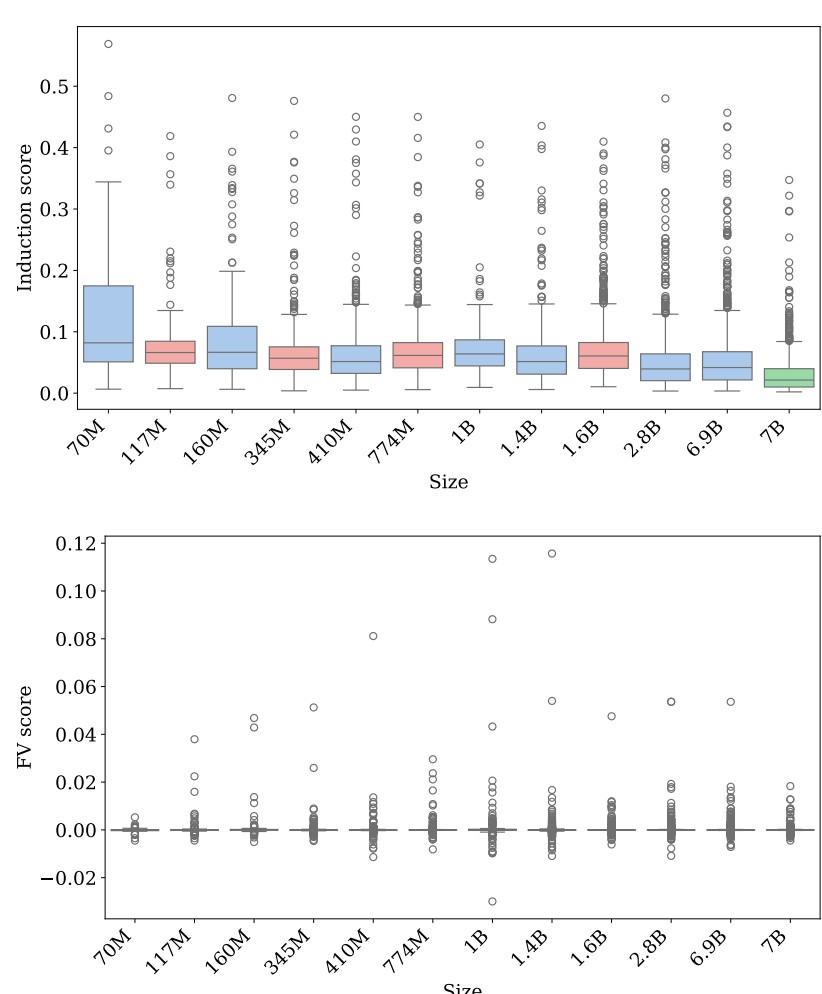

Figure 13: Distribution of induction scores (top) and FV scores (bottom) across model size.

### A.7 ICL TASKS

In Table 3, we list the ICL tasks used in this study. We refer to (Todd et al., 2024) and (Feng & Steinhardt, 2024) for a detailed description of each task.

### A.8 ABLATIONS BY TASK

In Figures 15-18, we plot the ICL accuracy after ablating induction heads and FV heads for each task in the evaluation set. We also compute the random baseline for each task, where we randomly sample outputs seen during training and compare these random outputs to the ground truth. The random baselines are shown in red horizontal lines.

### A.9 HEAD LOCATIONS

In Figure 19, we plot the locations of induction heads and FV heads across model layers.

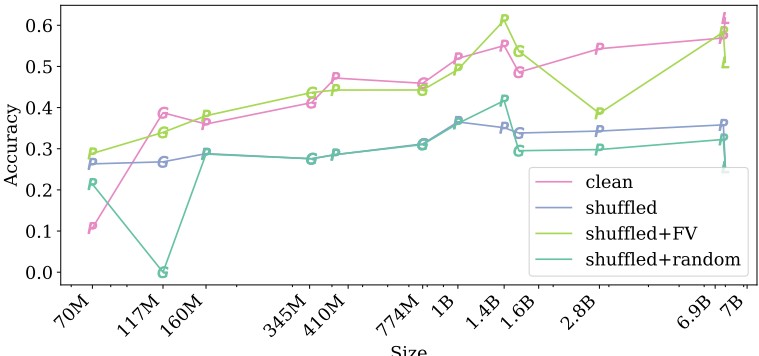

Figure 14: Model ICL accuracy on prompts with 10 in-context examples (clean), on uninformative shuffled prompts, on shuffled prompts with FV, and on shuffled prompts with random head outputs. Adding FV recovers most of the model accuracy on a clean run, with the exception of Pythia 2.8B.

## A.10    OVERLAP BETWEEN ABLATED INDUCTION AND FV HEADS

In Figure 20, we plot the percentage of attention heads that overlap between the set of induction heads and FV heads we ablate. We find that as the number of ablated heads increases, the overlap between the two sets of ablated heads also increases. This demonstrates the importance of performing ablations with exclusion to control for overlap.

## A.11    SCORES ACROSS TRAINING

In Figure 21, we plot the evolution of induction and FV scores averaged over top 2% heads across model training. In Figure 22, we plot the evolution of induction and FV scores of individual heads across training.

Table 3: Summary of ICL tasks used in our study. Tasks in **bold** are new tasks that were not used in (Todd et al., 2024).

| Task Name | Task Source |
| --- | --- |
| **Abstractive Tasks** | |
| **Abstract clf** | |
| Antonym | (Nguyen et al., 2017) |
| **Binding capital** | (Feng & Steinhardt, 2024) |
| **Binding capital parallel** | (Feng & Steinhardt, 2024) |
| **Binding fruit** | (Feng & Steinhardt, 2024) |
| **Binding shape** | (Feng & Steinhardt, 2024) |
| Capitalize first letter | (Nguyen et al., 2017) |
| **Capitalize index** | |
| **Capitalize second letter** | |
| Capitalize | (Nguyen et al., 2017) |
| Country-capital | (Todd et al., 2024) |
| Country-currency | (Todd et al., 2024) |
| English-French | (Conneau et al., 2017) |
| English-German | (Conneau et al., 2017) |
| English-Spanish | (Conneau et al., 2017) |
| **French-English** | (Conneau et al., 2017) |
| Landmark-Country | (Hernandez et al., 2024) |
| Lowercase first letter | (Todd et al., 2024) |
| National parks | (Todd et al., 2024) |
| Next-item | (Todd et al., 2024) |
| Previous-item | (Todd et al., 2024) |
| Park-country | (Todd et al., 2024) |
| Person-instrument | (Hernandez et al., 2024) |
| Person-occupation | (Hernandez et al., 2024) |
| Person-sport | (Hernandez et al., 2024) |
| Present-past | (Todd et al., 2024) |
| Product-company | (Hernandez et al., 2024) |
| Singular-plural | (Todd et al., 2024) |
| Synonym | (Nguyen et al., 2017) |
| CommonsenseQA (MC-QA) | (Talmor et al., 2019) |
| Sentiment analysis (SST-2) | (Socher et al., 2013) |
| AG News | (Zhang et al., 2015) |
| **Extractive Tasks** | |
| Adjective vs. verb | (Todd et al., 2024) |
| Animal vs. object | (Todd et al., 2024) |
| Choose first of list | (Todd et al., 2024) |
| Choose middle of list | (Todd et al., 2024) |
| Choose last of list | (Todd et al., 2024) |
| Color vs. animal | (Todd et al., 2024) |
| Concept vs. object | (Todd et al., 2024) |
| Fruit vs. animal | (Todd et al., 2024) |
| Object vs. concept | (Todd et al., 2024) |
| Verb vs. adjective | (Todd et al., 2024) |
| CoNLL-2003, NER-person | (Tjong Kim Sang & De Meulder, 2003) |
| CoNLL-2003, NER-location | (Tjong Kim Sang & De Meulder, 2003) |
| CoNLL-2003, NER-organization | (Tjong Kim Sang & De Meulder, 2003) |

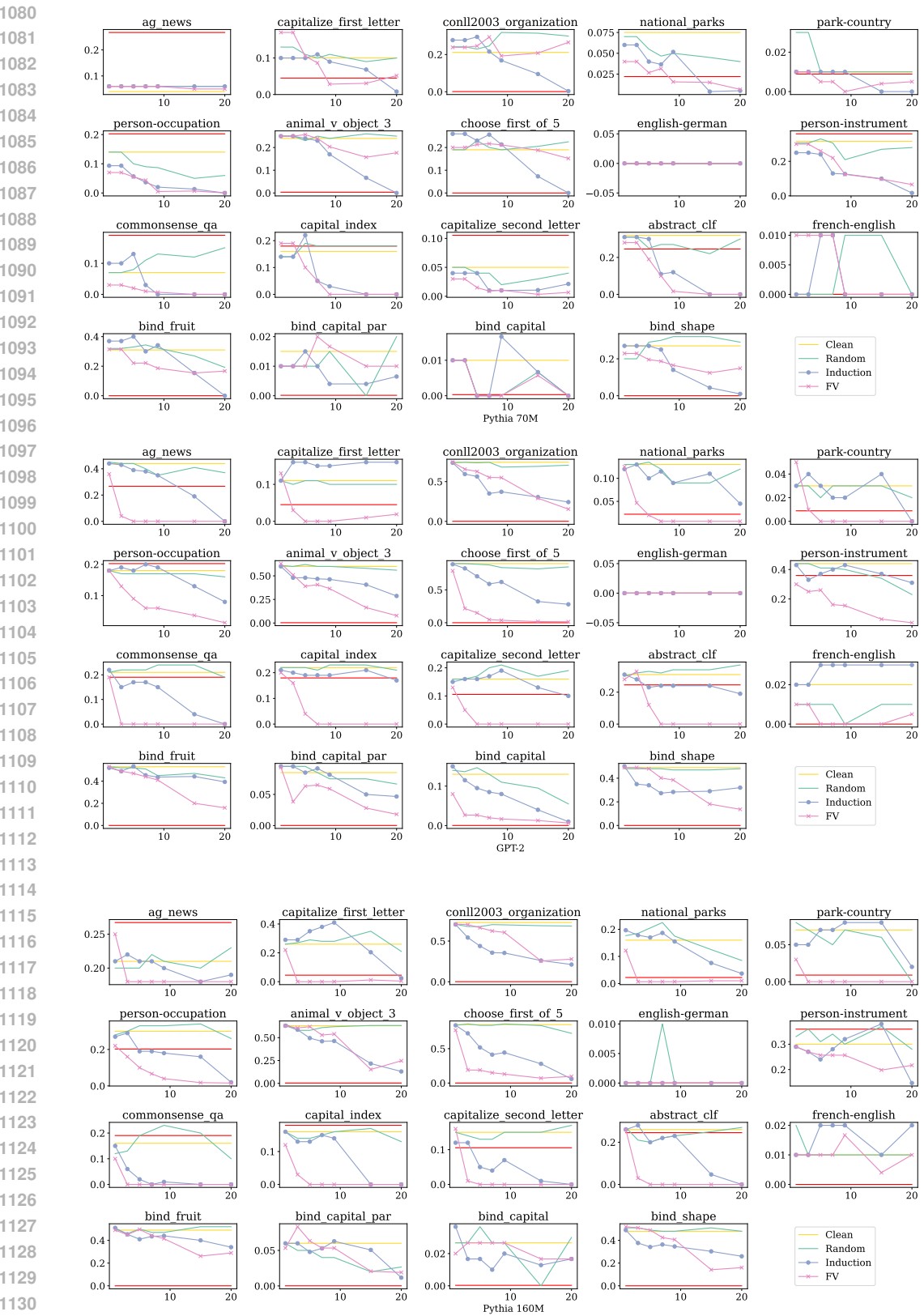

Figure 15: ICL accuracy after ablations by task. The red horizontal line represents the random baseline.

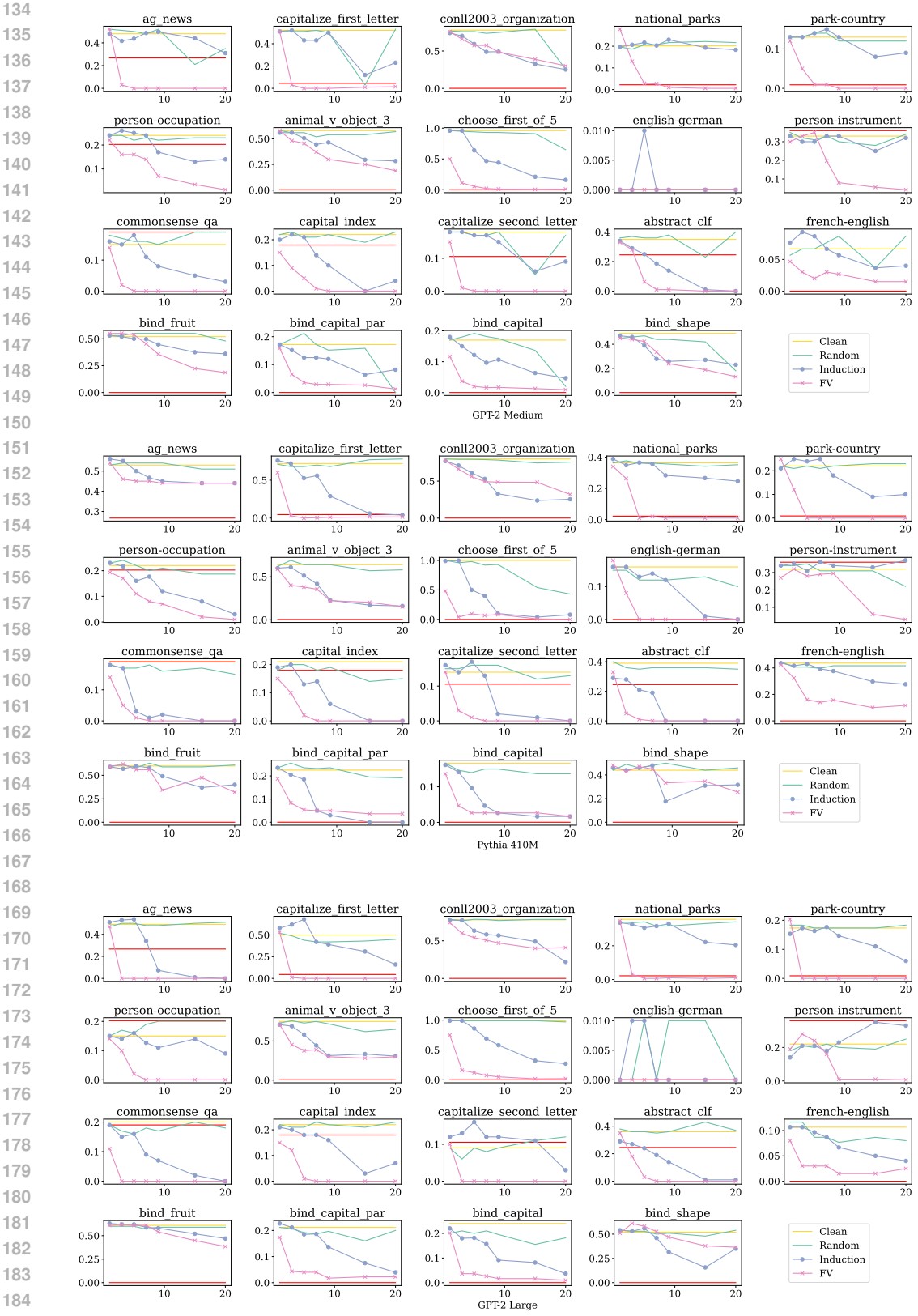

Figure 16: ICL accuracy after ablations by task. The red horizontal line represents the random baseline.

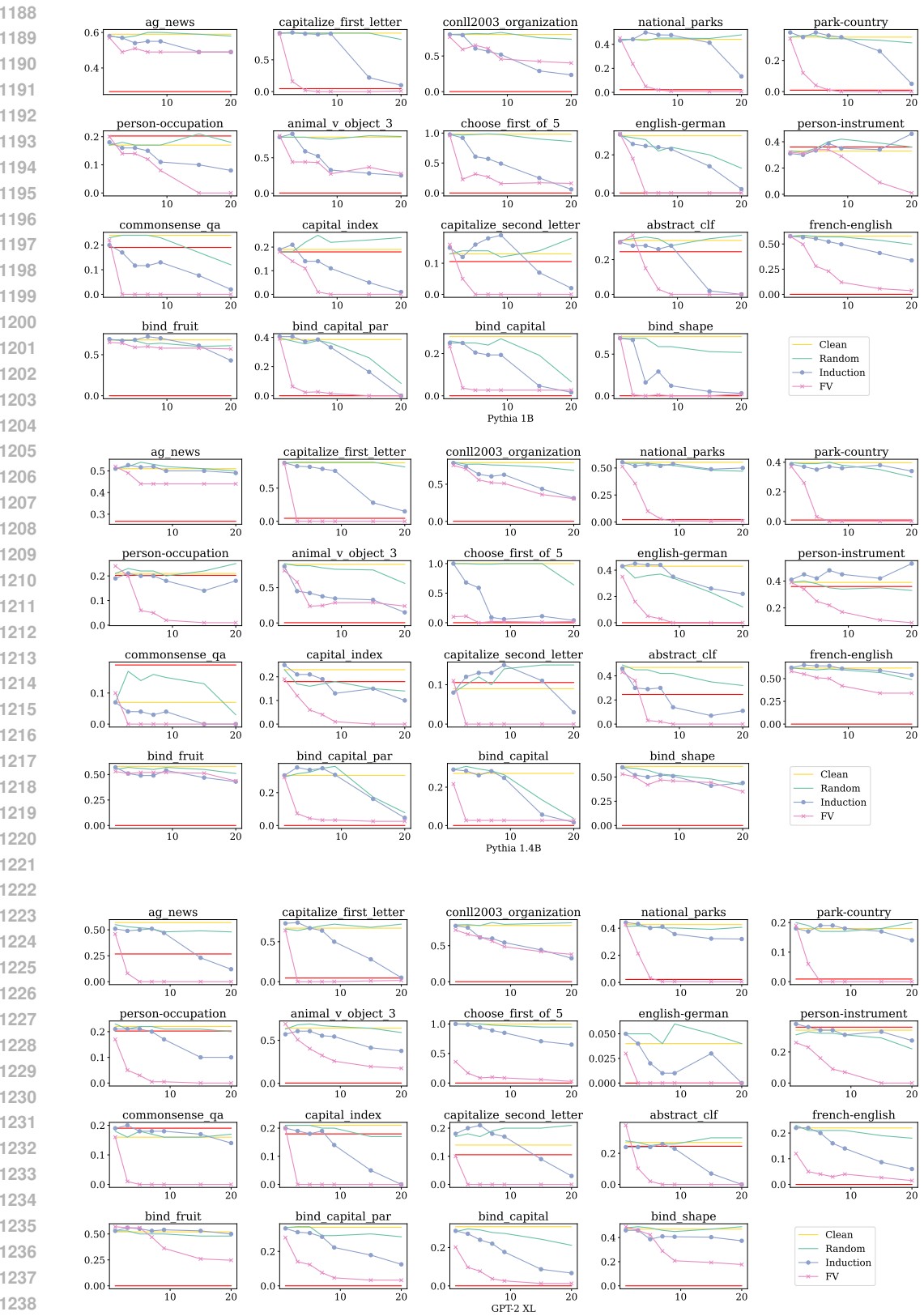

Figure 17: ICL accuracy after ablations by task. The red horizontal line represents the random baseline.

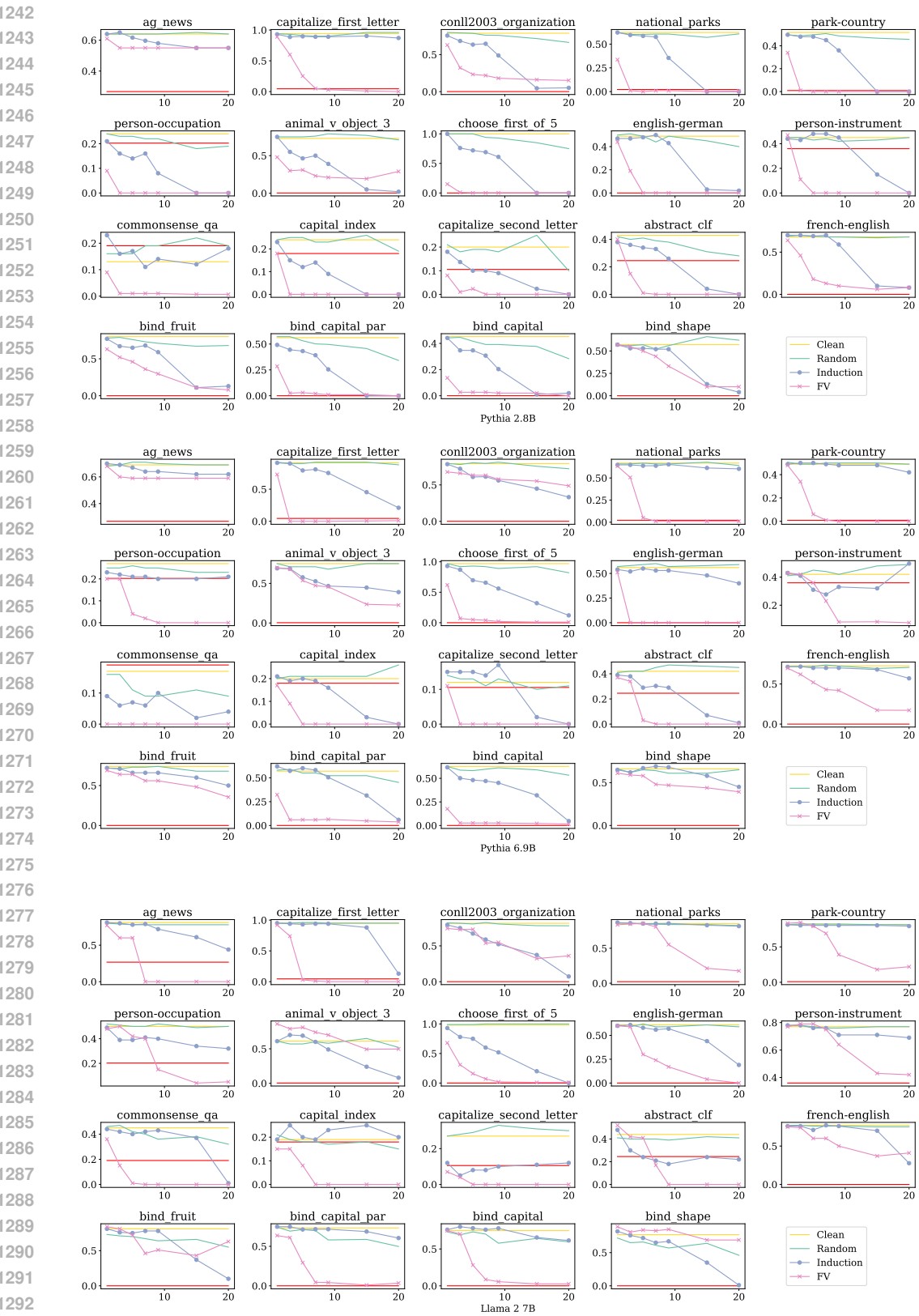

Figure 18: ICL accuracy after ablations by task. The red horizontal line represents the random baseline.

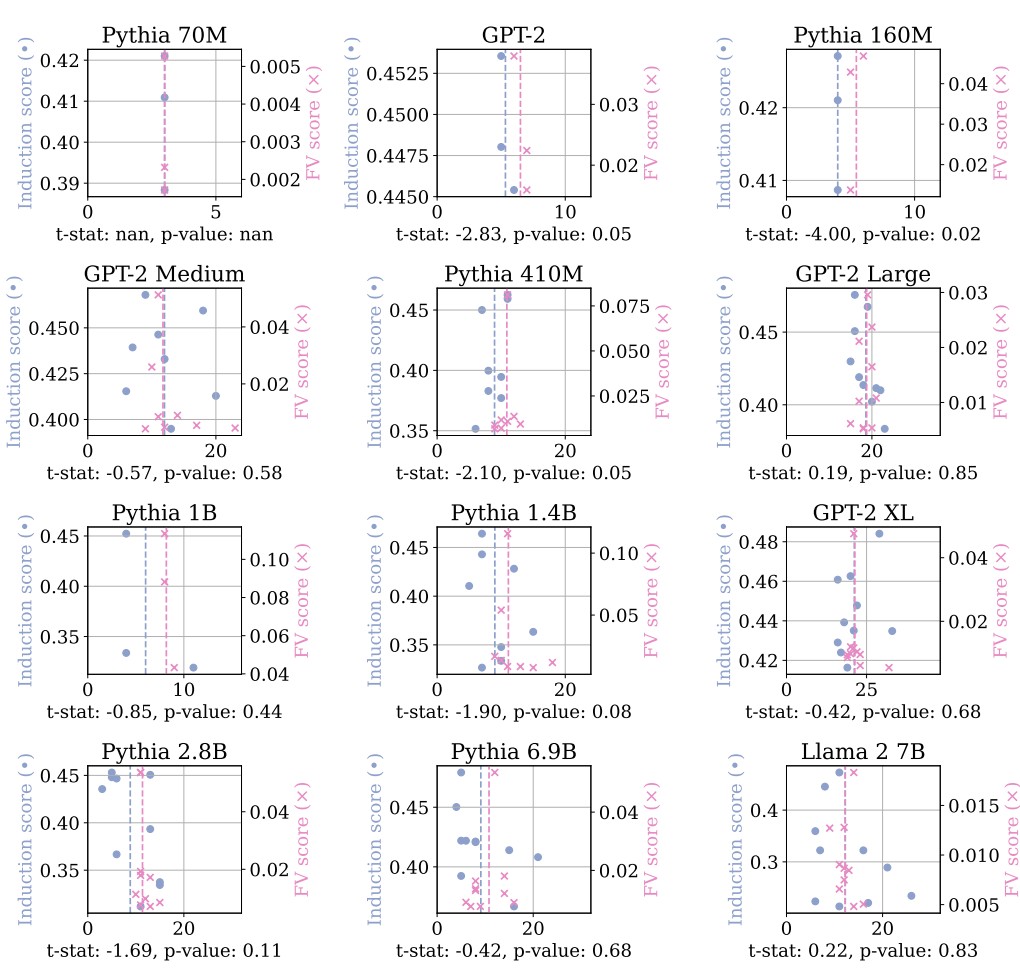

Figure 19: Location of induction heads (blue) and FV heads (pink) in model layers

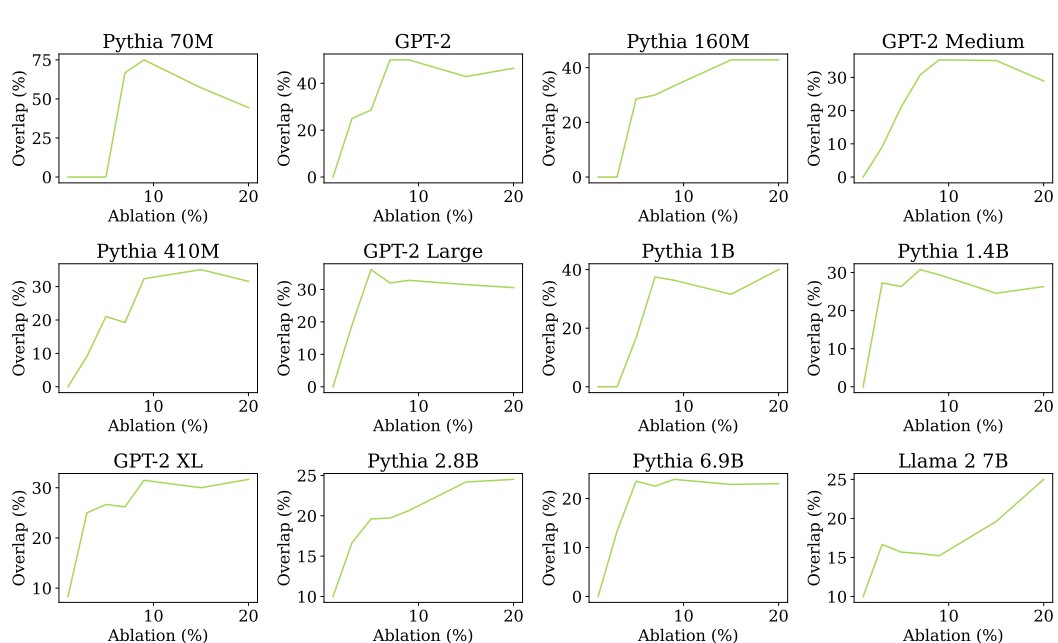

Figure 20: Overlap between set of induction heads and FV heads ablated.

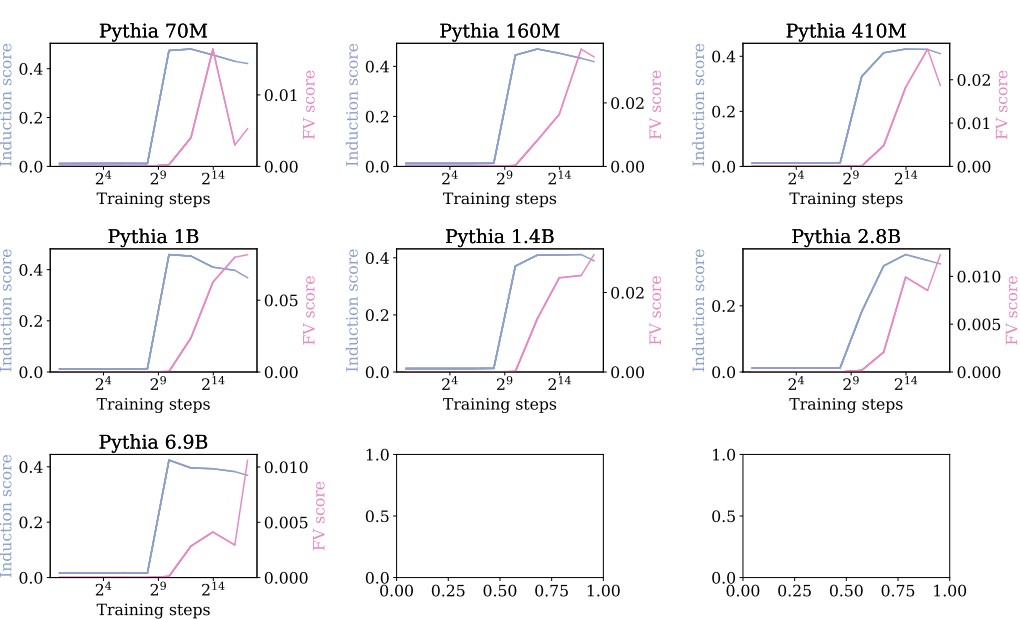

Figure 21: Evolution of induction score and FV score averaged over top 2% heads across training.

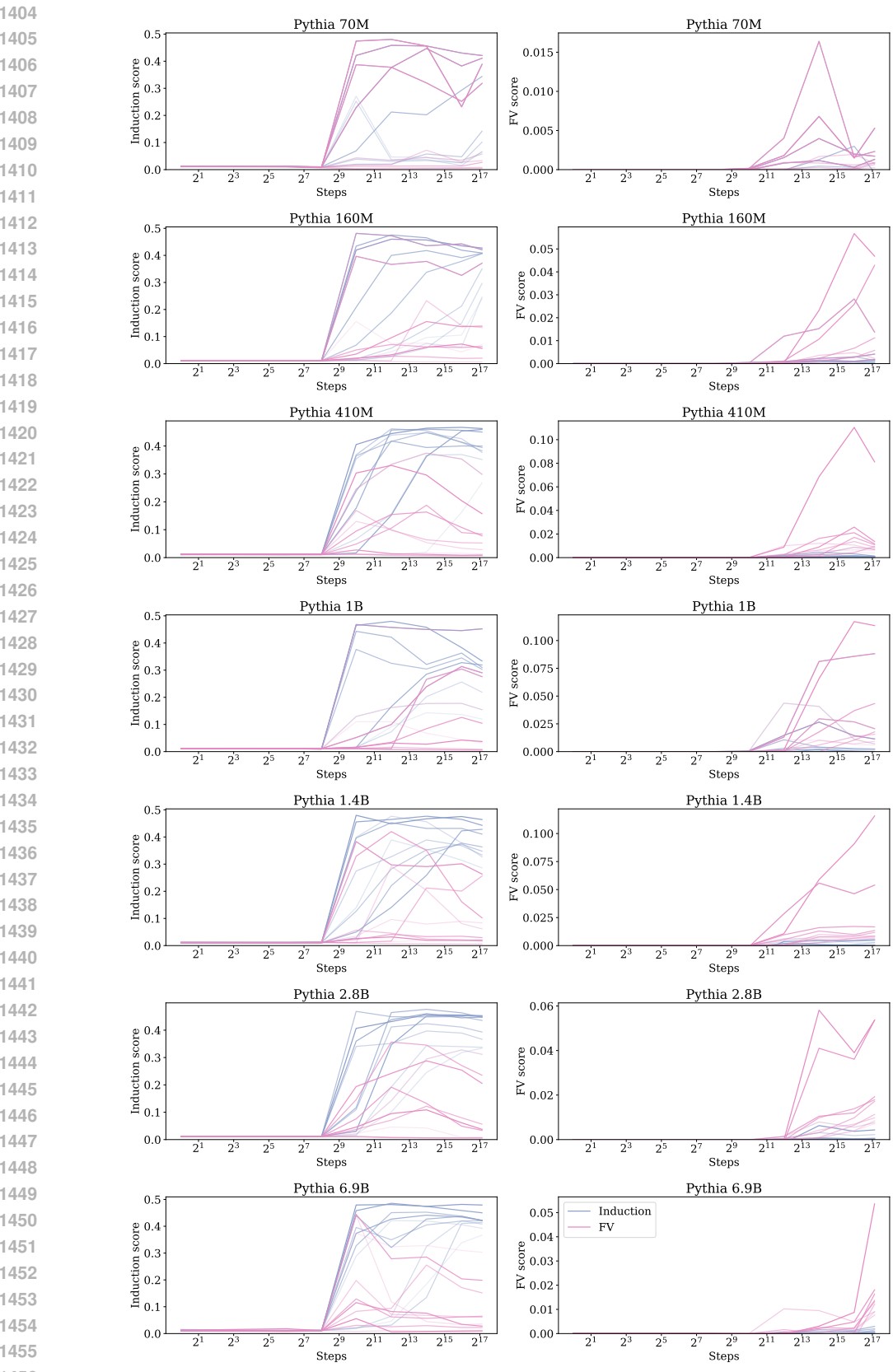

Figure 22: Evolution of induction scores (left) and FV scores (right) of individual induction and FV heads across training

