# OpenReview forum: "Which Attention Heads Matter for In-Context Learning?"
_ICLR.cc/2025/Conference — Submitted to ICLR 2025_

### Official Review · Reviewer_Ft6Y · 2024-10-28

**Soundness:** 3
**Presentation:** 3
**Contribution:** 3
**Rating:** 6
**Confidence:** 4

**Summary:**

This paper investigates the importance of attention heads in large language models (LLMs) under the lens of in-context learning (ICL). The paper compares "induction heads" from [1,2] and "FV heads" from [3], and finds that these two kinds of heads are distinct, but correlated. They show that reconciling different definitions of ICL help explain the difference.

___
[1] Elhage, et al. A Mathematical Framework for Transformer Circuits. 2021. (https://transformer-circuits.pub/2021/framework/index.html)

[2] Olsson, et al. In-Context Learning and Induction Heads. 2022 (https://transformer-circuits.pub/2022/in-context-learning-and-induction-heads/index.html)

[3] Todd, et al. Function Vectors in Large Language Models. 2023. (https://openreview.net/forum?id=AwyxtyMwaG)

**Strengths:**

- The writing of the paper structure was good and easy to follow
- The paper studied multiple model sizes and families to investigate the generality of the findings.

- Investigating induction heads more deeply in the few-shot ICL setting, and in larger models is interesting and worthwhile. While Olsson, et al. [2] does have a discussion about their reasoning for choosing token-loss difference, this paper suggests that different metrics (few-shot ICL vs. token-loss difference ICL) capture distinct effects. This trend is clear and consistent across models, suggesting that induction heads might not be the only contributors to (few-shot) ICL performance in language models.

**Weaknesses:**

- One of the main concerns I have with the results in this paper is that many of the models studied do poorly on few-shot ICL. For example,  Figure 4 and Figure 7 show that many of the small models average 20-30% accuracy across ~20 ICL tasks (this seems not great). If the models can't really do the task w/ few-shot prompting, how much can we really say about the "mechanism" behind it. In Line 318-319, it says: "We also plot ablations for all models, and ICL accuracy broken down by task, in Appendix A.2.", but as far as I could tell the breakdown of accuracy by task is missing. Can the authors clarify if the models they're testing are consistently better than baselines for the tasks they use (i.e. can the LMs "do" the task they're using to evaluate their claims)?

- The bulk of the evidence of "importance for attention heads" is based on mean ablation of attention heads. In some cases, ablation can cause adaptive computation (see McGrath, et al. [4]), and I wonder if there are other ways to verify that induction heads are not "important" for few-shot prompts, or FV heads are not "important" for token-loss difference other than ablation?

- In their paper, Olsson, et al. [2] do indicate that some induction heads they found that implement more sophisticated pattern matching while also fulfilling the traditional role of induction heads. It seems to me that the authors of [2] were aware that induction heads were not the entire story of ICL, but did not have words to describe it yet (e.g. they call this behavior "spiritually similar" to copying). I think this paper's Conjecture 1 matches this sentiment, and posit that perhaps FV heads are a generalization of induction heads, but I'm not sure whether this work engages with this previous acknowledgement by [2].
- Related to this -- While the paper does acknowledge the discrepancy in definitions of ICL in the literature (between few-shot vs. token-loss difference), some of the claims in the paper are a bit misleading for this exact same reason. For example:
   - Lines 51-52:  "This leads us to conclude that FV heads are mainly responsible for ICL, contrary to the prevailing belief that induction heads are a primary mechanism of ICL"
   - Lines 473-474: "Contrary to the prevailing consensus that ICL is largely driven by induction heads, we find that this assumption does not hold in most of the models we study."

The reason these types of statements misleading is because the experiments in Figure 4 and 7 of this paper still suggest that induction heads are important for the token-loss difference version of ICL. A way to make these kind of statements less misleading would be to qualify that this means "few-shot ICL" in the same sentence rather than a few sentences later on (e.g. "This leads us to conclude that FV heads are mainly responsible for **few-shot** ICL, ...", or "Contrary to the prevailing consensus that **few-shot** ICL is largely ...").
___

Minor Note:

- Many of the figures in the paper are averaged across tasks (e.g. Figure 1a, Figure 4, Appendix), but give no sense of variation of measurement. Adding confidence intervals or error bars would help provide more information about the range of the results presented.

___
[4] McGrath, et al. The Hydra Effect: Emergent Self-repair in Language Model Computations. 2023. (https://arxiv.org/abs/2307.15771)

**Questions:**

- In some models, some FV heads are also induction heads (which was also pointed out in Todd, et al [3], Appendix H. How do you feel this strengthens or contradicts the claims in the paper that induction heads are not important for few-shot ICL?

- In Figures 5 and 6, there is a nice general trend, but it's hard to tell which heads are correlated. For example, is the sharp spike in Pythia 6.9B (bottom right) the same FV head that has a induction score spike at around $2^{9}$?

---

> ### Author Response · Authors · 2024-12-02
>
> Thank you for this thorough and constructive review! Your detailed feedback will definitely help improve our paper.
>
> * We agree that we need to better contextualize our findings about ICL performance. In Appendix A.8, we have added the ablations broken down by task for each model, as well as a random baseline (red horizontal line) where we randomly sampled outputs from the training split to make predictions on the eval split. Most models, with the exception of Pythia 70M and 160M, outperform the random baseline. However, we also do note that by design, the random baseline is close to 0 for open-ended tasks instead of a finite set of labels. We have also computed majority class baselines (which by design, is similar to the random baseline we used) and zero-shot baselines using the same models we evaluate, the majority class baselines have similar performance to random baselines and zero-shot baselines are weaker. Since most of our models do show robust ICL abilities according to this evaluation, we believe it does not change the findings of the paper. Would you prefer we omit the smallest models with weak ICL abilities from our analysis, or retain the full spectrum while clarifying that the weakest models may not be able to do all the ICL tasks we evaluate on?
>
> * Thank you for raising the important point about adaptive computation! Taking this into account, the most direct approach would be to measure the extent of the hydra effect in our ablations. However, this is an open research problem since we ablate specific heads that score highly on induction or FV scores from different layers, whereas McGrath et al.'s method applies to ablating all heads in a single layer. Another approach would be to apply logit lens to measure the direct contributions of different head types to the output, but this will not account for second order effects either. We agree that ablation on its own may not sufficient to show something is *not* important, although if the hydra effect is present in both ablations of induction and FV heads to the same magnitude (not sure if this is the case), our findings from this comparison would remain valid. We will present our findings more carefully taking this into account, add a discussion about adaptive computation, and provide the approaches mentioned above as suggestions for future research.
>
> * We acknowledge Olsson et al.'s earlier conjecture that induction heads might be performing a more abstract version of the pattern matching-copying mechanism for ICL! However, we'd like to clarify that our C1 posits something different: FV heads are a distinct mechanism, not a generalization, of induction, but the existence of induction heads helps models learn the FV mechanism. This conjecture is based on results where some FV heads have high induction scores during training but get a drop in induction score while they increase in FV scores, while if FV is a generalization of induction, FV heads would maintain a high induction score.
>
> * Thank you for your point on potentially misleading claims! We'd like to maintain clarity in our claims, and we have updated "ICL" to "few-shot ICL" where that is the case. Please let us know if any misleading claims remain! Thank you for your minor note as well, we will add error bars where needed.
>
> Q1: We believe that the existence of FV heads that are also induction heads helps strengthen our claims and explain why previous literature attributed ICL to IH. Because of these heads, our work and prior works found that ablating IH (including FV heads that are also IH) leads to drops in ICL accuracy. However, once we only ablate non-FV IH, ICL accuracy is not affected as much. These heads also provide additional observations for C1, where FV heads "evolve" from IH, and the FV-IH heads are heads that did not fully lose induction mechanisms during training.
>
> Q2: Individual heads are identifiable by how saturated their curve is in both top and bottom plots, so the FV head spiking the most in bottom right is the most saturated (it has the highest final FV score) but this head has a weak spike in induction score (spikes at around 0.8 score at 2^9). If this is difficult to parse, we can add markers on the plots to differentiate each individual heads across plots.
>
> We apologize for our delayed response, and thank you again for helping us improve the clarity and rigor of our work. Please let us know if you have any remaining questions!

---

> > ### Comment · Reviewer_Ft6Y · 2024-12-03
> >
> > Thank you for the response, I appreciate the thorough response to my questions/concerns, and think many of them have been addressed.  With regard to the capabilities of small models, I think it's fine to include them with the clarification that they can't do all the ICL tasks used for evaluation. I think the proposed changes have strengthened the paper, and have helped scope/clarify the claims a bit more so I have updated my score to reflect this.

---

> > > ### Author Response · Authors · 2024-12-04
> > >
> > > Thank you for your reply and for updating the score, we appreciate it! We also agree with your comment on the small models, and have added the clarification to our draft.

---

### Official Review · Reviewer_rwqn · 2024-11-02

**Soundness:** 3
**Presentation:** 3
**Contribution:** 2
**Rating:** 3
**Confidence:** 4

**Summary:**

This paper examines the two primary existing explanations for the mechanisms behind in-context learning (ICL) – induction head and function vector (FV) head. First, the authors reveal the correlation between the head types. They find little head overlapping but a FV head usually has a relatively high induction score, and vice versa. Moreover, some of FV heads evolved from induction heads during training. Second, ablation experiments show FV head plays a more important role than induction head in ICL when using few-shot learning accuracy as the metric. The authors argue that this discrepancy from previous literature stems from the different metrics. Induction head was measured by token-loss difference rather than accuracy.

**Strengths:**

1. This paper presents a detailed empirical analysis and links the two ICL mechanisms. The mechanism of ICL is important for the development of LLMs.
2. This paper reveals that FV heads have a stronger causal effect on ICL performance than induction heads.
3. Experiments are solid.

**Weaknesses:**

1. This paper only makes comparisons between induction heads and FV heads, without any technical or theoretical improvements, nor providing a more effective explanation of the ICL mechanism. The indicators (induction score and function vector score) are borrowed from each original paper. So, the novelty is limited.
2. Although the authors present several new findings about induction and FV heads, it is still unclear how these findings will benefit future research works.

**Questions:**

Please discuss the contributions of your findings to promote the performance of current In-Context Learning in details.

---

> ### Author Response · Authors · 2024-12-02
>
> Thank you for raising this thoughtful point about novelty! We respectfully disagree, as our work provides several novel contributions beyond just comparison. Our joint analysis reveals previously unknown interactions between FV and induction heads, providing a more complete picture of ICL mechanisms. We also perform several novel analyses for the study of ICL mechanisms (such as performing ablations of one head type while excluding another, tracking the evolution of FV and induction heads over training in both their induction and FV scores). These findings help resolve apparent misunderstandings in prior work about induction heads' role in ICL.
> By bridging these previously separate lines of research, we advance our theoretical understanding of ICL and encourage future interpretability research to further study FV heads when studying ICL. We appreciate the opportunity to clarify these aspects of our work and thank you for helping us improve its presentation!
>
> Thank you as well for noting the impact on future work! Our findings have direct implications for future research: in mechanistic understanding (clarifying how FV heads emerge during training, identifying which heads drive few-shot vs. token-loss ICL performance), and in recommended practices for interpretability research more generally (shift focus from induction heads to FV heads when studying ICL, question the universality hypothesis of interpretability, establish that different definitions of ICL in the literature measure very different things). These contributions provide concrete directions for improving model understanding. We'll clarify these research implications in the revision!

---

### Official Review · Reviewer_wGjS · 2024-11-03

**Soundness:** 2
**Presentation:** 3
**Contribution:** 2
**Rating:** 3
**Confidence:** 3

**Summary:**

The paper compares the phenomena of induction heads (Olsson et al., 2022) and Function Vector heads (FV heads; Todd et al., 2024), both of which are key attention heads to in-context learning shown in previous interpretability literature, across 12 language models. The paper finds that the set of induction heads and FV heads are mostly distinct and FV heads usually appear deeper in models than induction heads, but there are correlations between the induction scores and the FV scores of the top induction heads and top FV heads, showing a correlation between these two distinct sets. Moreover, the paper examines the training dynamics of the selected models, and finds that FV heads are learned later in LMs than induction heads with respect to training steps and some induction heads become FV heads through training. Finally, the paper proposes two competing hypotheses to explain such differences and similarities between induction heads and FV heads, leaving a further investigation for future work.

**Strengths:**

1. The paper investigates two influential ideas in LLM interpretability that aim at explaining in-context learning, and is very well-motivated.

2. The paper conducts thorough empirical investigations between induction heads on a wide range of language models and FV heads and sheds light on a better understanding of how large language models learn and implement in-context learning.

3. To the best of my knowledge, the paper is the first to explore how induction/FV heads are learned and formed during pre-training. This is in my opinion a concrete contribution to the community.

4. The paper is well-written with good presentations.

**Weaknesses:**

1. One of the key findings of the paper is that FV heads appear in deeper layers than induction heads do. However, by inspecting Figure 2 and Figure 13, I think the average layers of FV heads and induction heads do not look very far from each other (most of them differ by 1-2 layers). Particularly, the average layers for GPT2 Medium, GPT2 Large, GPT-2XL, and Llama 2-7B models seem to be the same. Therefore, I think some forms of statistical tests might need to be done here to strengthen the argument.

2. Another argument the paper makes is that induction heads and FV heads are distinct using the metric defined in line 260. However, it is possible that neighboring attention heads could be performing similar functionalities in LLMs, as shown by [1] and follow-up works. I think only measuring the exact overlap between the two sets of heads might be a bit misleading; it might be better to measure the overlap of layers where the two sets of heads reside. For example, the Pythia 6.9 B plot in Figure 2 shows that both sets have some heads in layer 8 and layer 17.

3. The paper claims that "previous studies on induction heads focus on small model sizes" (line 187). However, I think this is inaccurate as works such as [2][3] have already extended induction heads to large models up to 20B/66B. Moreover, these papers also discuss the effects of scales on induction heads and in-context learning of LLMs. These prior works are not discussed in the paper, making the paper's claim of contribution for investigating induction heads at larger scales weaker.

4. Combining 1-3, I think some of the main findings of the paper might be a bit fragile and relatively incremental.

[1] Thomas McGrath, Matthew Rahtz, Janos Kramar, Vladimir Mikulik, and Shane Legg. The Hydra Effect: Emergent Self-repair in Language Model Computations. 2023
[2] Joy Crosbie and Ekaterina Shutova. Induction Heads as an Essential Mechanism for Pattern Matching in In-context Learning. 2024
[3] Hritik Bansal, Karthik Gopalakrishnan, Saket Dingliwal, Sravan Bodapati, Katrin Kirchhoff, and Dan Roth. Rethinking the Role of Scale for In-Context Learning: An Interpretability-based Case Study at 66 Billion Scale. 2023

**Questions:**

1. For the induction heads and FV heads analyzed in Section 5, are they obtained from each checkpoint separately, or they are obtained from the last checkpoint only?

2. Regarding conjecture C2 discussed in Section 6, why would it "predict that ablating monosemantic FV heads would not hurt ICL performance? (line 466)? Is it possible that the monosemantic FV heads are important task-specific operations beyond copying (which is the mechanism implemented by induction heads)?

---

> ### Author Response · Authors · 2024-12-02
>
> Thank you for this detailed and constructive review! You raise several important points that will help strengthen our paper. To start, since you cited the layer location and overlap of heads as our key findings, we would like to clarify that our key findings are those illustrated in Figure 1: (1) ablating FV heads affects ICL score more than ablating induction heads, and (2) many FV heads evolve from induction heads during training. We present results on the layer location and head overlap primarily as a precursor to these findings, to first establish that FV and induction heads are indeed two distinct types of heads and therefore should be analyzed separately. We will clarify this in our paper!
>
> W1: Thank you for raising this important point! We have added t-stat and p-values for the difference in head layer locations for each model in Figure 19 (Appendix A.9). As you have pointed out, the mean layer is often the same in many of our models, so we will clarify that this result does not hold in all models, but when the mean layer does differ, FV head mean layer is always deeper than IH mean layer. We will also emphasize the next result on head overlap more to establish the distinctness of the two types of heads.
>
> W2: You raised that neighboring/other attention heads may perform similar functionalities to FV and induction heads which we might miss if we take exact overlap. Our FV score and induction score metrics precisely measure how much heads perform these functionalities, which is why we use these metrics to construct the set of heads to compare (e.g. these metrics would exclude heads in the same layer as FV heads but that does not perform FV functionalities). Also, please let us know if we misunderstood your point, since you cited McGrath et al. [1], however we understand this work as when we ablate all heads from a layer, a subsequent layer performs the functionalities of the ablated layer. Our section on head overlap does not perform any ablations yet so there would be no self-repair effects.
>
> W3: Thank you for bringing the two related works to our attention, we have added them to our discussion! We'd also like to clarify that in our section on reconciling divergent findings, it stems from either because small models do rely on induction heads, *or* because earlier studies (on small or large models) do not account for the correlation of FV and induction heads. We believe the two works you have provided fall into the second category, where studying only induction head ablations can give an incomplete account of the ICL mechanism landscape. Our work reproduces these two works' findings as well, and we'd also like to clarify that we did not intend to claim to be the first to study ICL heads at scale. Please let us know if you found any of our claims misleading!
>
> Q1: We calculated the induction and FV scores of attention heads from each checkpoint separately to compute the evolution of scores for specific heads.
>
> Q2: C2 states that the FV heads used for ICL are polysemantic heads that implement a combination of induction and a different mechanism. Therefore, monosemantic FV heads that do not perform induction mechanisms are excluded from this category of ICL heads, which is why C2 would predict that ablating these heads would not hurt performance. We agree that since we do observe ablating non-induction FV heads drops ICL score, these monosemantic may be performing important operations beyond induction, hence why we state that this observation contradicts a prediction from C2.
>
> We apologize for our delayed response, and thank you again for helping us improve the clarity and rigor of our work. Please let us know if you have any remaining questions!

---

### Official Review · Reviewer_1PLt · 2024-11-04

**Soundness:** 2
**Presentation:** 2
**Contribution:** 2
**Rating:** 3
**Confidence:** 3

**Summary:**

The paper studies the role of different attention heads to LLMs' in-context learning (ICL) capability. Specifically, the paper studies two types of attention heads, induction heads and function vector (FV) heads. Through empirical experiments, as opposed to prior belief, the authors find that ICL is mainly driven by FV heads. In addition, there is potential connection between induction heads and FV heads.

**Strengths:**

- It is interesting to understand ICL capability by connecting it to attention heads with certain mechanisms.
- I appreciate the authors disambiguate ICL and token-loss difference, which allows further disentanglement between the effect of induction heads and FV heads.
- I like the controlled ablation approach to separate the effect of induction heads and FV heads.

**Weaknesses:**

- I am not very convinced by the ablation method used in section 4.1, i.e., by replacing output vector by mean values. It seems a bit ad-hoc for me without further justification. Why use mean but not other statistics? How robust are the results, or is it specific only to the ablation method used here?
- Given that induction heads and FV heads appear at different locations (layers) within the model, head "location" can be one confounding factor that contributes to the difference in ICL performance when ablating induction heads vs. FV heads. There should perhaps be a controlled baseline that ablates heads at different locations in the model.
- The empirical results presented in the paper appear a bit weak. It is not clear how many tasks are evaluated (Is Figure 4 showing averaged results?), and which ICL tasks are used exactly? How well do the tasks represent real-world ICL/few-shot use cases?
- Some conclusions made from the observations seem more like conjectures instead of actual proof. Paper can be made more sound to clarify conjectures from conclusions with substantiated results. E.g., Line 252: "This suggests that induction and FV heads may not fully overlap, and that FV heads may implement more complex or abstract computations than induction heads".
- Minor: The paper presentation can be improved with clearer background introduction of induction heads and FV heads.

**Questions:**

- Sec 3.2 results are a bit confusing. If induction heads and FV heads are distinct (not overlapping), how could FV heads also have high induction scores, or vice versa? Does it suggest that there are some overlapping heads that have both high induction and FV scores?
- While it is interesting to understand ICL by connecting it to certain model attention heads, what are some actionable improvements/implications we could make after establishing the connection?

**Details Of Ethics Concerns:**

No immediate ethics concerns.

---

> ### Author Response · Authors · 2024-12-02
>
> Thank you for this detailed and thoughtful review. You raise several important points that will help improve our paper!
>
> W1: Great question regarding our choice of ablation method! We chose mean ablation following recommendations from Hase et al., 2021 [1], Zhang and Nanda 2024 [2], Wang et al., 2022 [3] where the suggest using mean ablation over zero ablation to avoid the out-of-distribution problem with zero ablations. Since the Olsson et al., 2022 paper on induction heads use zero ablation in their ablation studies, we have run additional experiments using zero ablations (replacing the ablated head's output by a zero vector), as well as random ablations (replacing the ablated head's output by a random vector). We added these results in Appendix A.3 (Figure 9). Overall, these ablations give similar results to mean ablations and our findings are robust to the choice of ablation method.
>
> W2: Great point, thank you! We performed ablations of random heads in the different layers that correspond to where induction and FV heads appear in Pythia 6.9B, and present results in Appendix A.4 (Figure 10) of the revision. While ablating random heads in certain layers, namely the later layers, gives more drop to ICL score than other (earlier) layers, overall, the effect of ablating random heads is much weaker than ablating induction / FV heads. Therefore, we are no longer concerned about the confounding factors of head location in this comparison.
>
> W3: We provide a list of ICL tasks we use in Appendix A.7 (Table 3), and data for these tasks will also be provided with the code release. While some tasks would not reflect real-world use cases (e.g. "capitalize second letter", "next-item"), most of these tasks would represent real-world use cases (e.g. English-French lexical translation, sentiment analysis, currency of countries).
>
> W4: Thank you for your feedback on our clarity! We have revised Table 1 to better distinguish findings backed by evidence in our paper and conjectures with no sufficient evidence.
>
> W5: Thank you! We have added additional intuitive explanations of FV heads, please let us know if the background in Section 2 remains unclear.
>
> Q1: We chose the top 2% (98th percentile) based on induction/FV scores to define our sets of induction and FV heads in Section 3.2, and calculated the overlap between these sets. While most induction heads do not lie in the 98th percentile of FV heads and vice versa, the middle and right plots in Fig 3 show that induction heads still have relatively high FV scores (90-95th percentile) and vice versa. This result is especially important for ablation experiments later in the paper, where naively ablating induction heads would also ablate FV heads with high induction scores and vice versa, hence why we perform ablations with exclusion to account for the overlap during ablation. We hope this clarifies!
>
> Q2: For actionable implications, we believe that future work exploring how to inject or encourage smaller models to develop FV heads may be promising to improve ICL performance in small models. We would also be excited to see more work in the field of interpretability studying these FV heads, for example what exactly these function vectors are encoding or what mechanism FV heads are more concretely implementing, since a lot of works on ICL interpretability so far have focused on induction heads.
>
> We apologize for the delayed response, and we appreciate your feedback helping strengthen the experiments and presentation of our work!
>
> [1] Peter Hase, Harry Xie, and Mohit Bansal. The out-of-distribution problem in explainability and search methods for feature imporrtance explanations. Advances in neural information processing systems, 34:3650–3666, 2021.
> [2] Fred Zhang and Neel Nanda. Towards best practices of activation patching in language models: Metrics and methods. In The Twelfth International Conference on Learning Representations, 2024.
> [3] Kevin Ro Wang, Alexandre Variengien, Arthur Conmy, Buck Shlegeris, and Jacob Steinhardt. Interpretability in the wild: a circuit for indirect object identification in GPT-2 small. In The Eleventh International Conference on Learning Representations, 2023.

---

### Official Review · Reviewer_oNGh · 2024-11-04

**Soundness:** 3
**Presentation:** 4
**Contribution:** 3
**Rating:** 8
**Confidence:** 3

**Summary:**

This paper investigates the role of distinct types of attention heads—induction heads and function vector (FV) heads - in supporting ICL in LLMs. Using a set of ablation experiments across 12 transformer-based models, the authors find that FV heads are primarily responsible for effective ICL, particularly as model size increases. The study further explores the interplay between induction and FV heads, finding that many FV heads evolve from induction heads during training.

**Strengths:**

1. The study provides a fresh perspective on the mechanisms of ICL, highlighting the underestimated role of FV heads compared to induction heads. By introducing the idea that induction heads may serve as a precursor to FV heads, it opens new directions for understanding head evolution during model training.
2. The experiments are thorough, covering a range of model sizes and carefully controlled ablation studies. The authors demonstrate rigor in handling overlapping effects and using multiple model families (Pythia, GPT-2, and Llama 2) to validate findings.
3.  The paper is well-organized and clearly written. The authors provide a comprehensive background on induction and FV heads and lay out their methodologies and results in an accessible manner, making it easy to follow their conclusions.
4. This work addresses a fundamental question in the interpretability of language models and challenges existing beliefs about ICL. The findings contribute to a more comprehensive understanding of the roles of different attention mechanisms and could guide future work on interpretability techniques.

**Weaknesses:**

1. The conjecture that induction heads serve as a precursor to FV heads is supported by empirical observations but could be further validated by testing whether removing induction heads impacts the development of FV heads during training. This could help confirm the proposed causal relationship.

**Questions:**

1. Do the authors have insights into why the Llama 2 model exhibited lower FV scores, and how this finding might relate to differences in its architecture or training procedure compared to the other models?
2. Is there an explanation as to why the gap between the effect of FV heads and induction heads increases with model scale?

---

> ### Author Response · Authors · 2024-12-02
>
> Thank you for your thorough and enthusiastic feedback!
>
> Studying the effect of removing induction heads during training is a great suggestion! Although we do not have enough resources to re-train Pythia models with ablated heads, we agree that this experiment could help confirm C1, and we have added this suggestion for future work in our revision.
>
> To address your questions:
> 1. This is an interesting question! We find it very curious that although Llama 2 7B has low induction and FV scores relative to other models (Figure 12), in our ablation experiments, ablating FV heads still lead to strong drops in ICL accuracy. One possible explanation is that these FV heads are polysemantic, so they exhibit weaker FV functionalities but implement a third functionality from induction/FV that are also important to ICL performance. Pythia 6.9B and Llama 2 7B are comparable in size, number of layers, number of attention heads, and size of attention heads, so this rules out these factors as the properties responsible for this finding. We believe the main difference might be the type of attention head used (Flash Attention in Pythia vs. Grouped Query Attention in Llama 2), further experiments comparing these two attention head types would help confirm this difference!
>
> 2. As for the effect of FV heads and induction heads with scale, we hypothesize that because FV heads are more complex (appear in deeper layers and later during training), FV heads that effectively perform ICL only emerge with scale, whereas induction heads are simpler to learn but perform ICL less accurately. Therefore, in smaller models, the contributions of the less accurate induction heads and the under-developed FV heads are comparable, whereas in larger models, the well-trained FV heads contribute more to ICL than the simpler induction heads.
>
> We apologize for the delayed response, and thank you once again for your feedback!

---

### Author Response · Authors · 2024-12-02
**Summary of PDF revisions**

We thank all reviewers for their helpful and constructive feedback! We also apologize for the delayed responses due to personal challenges faced by our first author. While we understand this gives reviewers limited time to engage with our revisions, we would be deeply grateful if you could take a moment to read our responses where possible.

We are excited by this work since it significantly updates the community's understanding of attention mechanisms behind ICL, and explains why previous results linking induction heads to ICL paint an incomplete picture! We appreciate your time and dedication in helping improve this work!

Taking reviewer feedback into account, we have made the following updates to our revised PDF:
* Added random and zero ablation results to verify the robustness of our ablation experiments (Appendix A.3)
* Added ablations of random heads in specific layers to verify the confounding factor of head location in ablation experiments (Appendix A.4)
* Added random baselines to task-specific ICL performance of each model (Appendix A.8)
* Added statistical tests for layer locations of induction and FV heads (Appendix A.9)
* Clarify claims substantiated by findings from conjectures/hypotheses (Table 1)
* Clarify instances of few-shot ICL instead of general ICL (throughout the PDF)
* Added intuitive explanations of induction and FV heads (Section 2)
* Added and discussed missing references (Section 2.3)

We will also continue to make revisions based on recent discussions with reviewers. We hope our additional experiments in the revision and our responses help clarify our work and alleviate concerns. Given these changes, we would greatly appreciate it if reviewers consider revising their score!

---

### Author Response · Authors · 2024-12-04
**Summary of discussion phase (1/2)**

As the discussion phase ends, we thank all the reviewers for their thoughtful feedback and time!
During the discussion, reviewers recognized the **significant contributions** of our work:
## 1. Challenges existing beliefs of ICL with implications for future interpretability research
* "This work addresses a fundamental question in the interpretability of language models and **challenges existing beliefs about ICL** [...] **could guide future work** on interpretability techniques." (oNGh)
* "The study provides a fresh perspective on the mechanisms of ICL, **highlighting the underestimated role of FV heads** compared to induction heads." (oNGh)
* "The authors disambiguate ICL and token-loss difference, which allows further disentanglement between the effect of induction heads and FV heads" (1PLt)
* "This paper suggests that **different metrics (few-shot ICL vs. token-loss difference ICL) capture distinct effects**. This trend is clear and consistent across models, suggesting that induction heads might not be the only contributors to (few-shot) ICL performance in language models." (Ft6Y)
## 2. Novel analyses and understanding of ICL
* "By introducing the idea that induction heads may serve as a precursor to FV heads, it opens **new directions for understanding head evolution during model training**." (oNGh)
* "The paper is the **first to explore how induction/FV heads are learned and formed during pre-training**. This is in my opinion a concrete contribution to the community." (wGjS)
* "The authors present several new findings about induction and FV heads" (rwqn)
## 3. Rigor of experiments
* "The experiments are thorough, covering a range of model sizes and carefully controlled ablation studies. The authors demonstrate **rigor in handling overlapping effects** and using multiple model families (Pythia, GPT-2, and Llama 2) to validate findings." (oNGh)
* "I like the controlled ablation approach to separate the effect of induction heads and FV heads." (1PLt)
* "The paper conducts **thorough empirical investigations** between induction heads on a wide range of language models and FV heads and sheds light on a better understanding of how large language models learn and implement in-context learning" (wGjS)
* "The paper studied **multiple model sizes and families** to investigate the generality of the findings." (Ft6Y)
* "Experiments are solid." (rwqn)
## 4. Presentation clarity
* "The paper is well-organized and clearly written." (oNGh)
* "The paper is well-written with good presentations." (wGjS)
* "The writing of the paper structure was good and easy to follow" (Ft6Y)

---

> ### Author Response · Authors · 2024-12-04
> **Summary of discussion phase (2/2)**
>
> To our knowledge, we have also responded to all reviewer questions, clarified confusions, and **addressed all major concerns** reviewers brought up:
> ## 1. Questions about robustness of results
> * *"I am not very convinced by the ablation method used in section 4.1, i.e., by replacing output vector by mean values. [...] How robust are the results, or is it specific only to the ablation method used here?" (1PLt)*
>
> We provided references explaining the advantage of mean ablation over other ablation methods (zero/random). We also reproduced experiments using zero ablation and random ablation instead, and show that our **results are robust against ablation method**.
>
> * *"Head "location" can be one confounding factor that contributes to the difference in ICL performance [...]. There should perhaps be a controlled baseline that ablates heads at different locations in the model."  (1PLt)*
>
> We performed ablations of random heads at different layers as additional baselines. The effect of ablating random heads is minimal relative to ablating induction / FV heads, regardless of the layer chosen. We verified that **head location is not a confounding factor**.
>
> * *"Some forms of statistical tests might need to be done here to strengthen the argument [that FV heads appear in deeper layers than induction heads do]." (wGjS)*
>
> We added t-stat and p-values for the difference in head layer locations for each model in Figure 19 (Appendix A.9).
>
> * *"It is possible that neighboring attention heads could be performing similar functionalities [as FV and induction heads] in LLMs." (wGjS)*
>
> Our FV score and induction score metrics precisely measure whether each head performs these functionalities, so this is a non-issue.
>
> * *"Can the authors clarify if the models they're testing are consistently better than baselines for the tasks they use?" (Ft6Y)*
>
> We added a random baseline for each task we used. Most models, with the exception of Pythia 70M and 160M, **outperform the random baseline**, and clarified in our paper that these two small models can't do all the ICL tasks used.
>
>
>
> ## 2. Presentation
> * *"It is not clear how many tasks are evaluated, and which ICL tasks are used exactly? How well do the tasks represent real-world ICL/few-shot use cases?"  (1PLt)*
>
> We provide a list of ICL tasks we use in Appendix A.7 (Table 3). Most tasks represent real-world use cases.
>
> * *"Paper can be made more sound to clarify conjectures from conclusions with substantiated results"  (1PLt)*
>
> We revised Table 1 to better distinguish findings backed by evidence in our paper and conjectures with no sufficient evidence.
>
> * *"Works such as [2][3] have already extended induction heads to large models up to 20B/66B" (wGjS)*
>
> We added a discussion of [2][3] in our related work. These two works provide further examples of one of our paper's claims: studying only induction head ablations gives an **incomplete account** of the ICL mechanism landscape.
>
> * *"In some cases, ablation can cause adaptive computation [1]" (Ft6Y)*
>
> We added a discussion about adaptive computation and presented our conclusions more carefully taking this into account. We also suggested future research directions to take into account adaptive computation.
>
> * *"Some of the claims in the paper are a bit misleading [...] A way to make these kind of statements less misleading would be to qualify that this means "few-shot ICL" in the same sentence" (Ft6Y)*
>
> We updated "ICL" to "few-shot ICL" where that is the case to improve clarity.
>
> **We thank all reviewers for their helpful feedback! During the discussion phase, we significantly improved the robustness of our results with additional experiments, and further clarified our paper's claims.**
>
> We are excited by this work since it **challenges the community's existing beliefs** about the role of induction heads in ICL, and provides **important lessons/directions for future interpretability research** (disentangle "few-shot" vs. "token-loss difference" ICL, universality hypothesis in interpretability, conjectures on the evolution of FV heads from induction heads). We hope our work will also inspire more research studying FV heads in ICL!
>
> ------
>
> [1] Thomas McGrath, Matthew Rahtz, Janos Kramar, Vladimir Mikulik, and Shane Legg. The Hydra Effect: Emergent Self-repair in Language Model Computations. 2023 [2] Joy Crosbie and Ekaterina Shutova. Induction Heads as an Essential Mechanism for Pattern Matching in In-context Learning. 2024 [3] Hritik Bansal, Karthik Gopalakrishnan, Saket Dingliwal, Sravan Bodapati, Katrin Kirchhoff, and Dan Roth. Rethinking the Role of Scale for In-Context Learning: An Interpretability-based Case Study at 66 Billion Scale. 2023

---

### Meta-Review · Area_Chair_Dzjm · 2024-12-23

**Metareview:**

While the paper provides interesting insights into the roles of induction heads and function vector heads in in-context learning, several key concerns remain. First, some findings are presented as conclusions but lack sufficient empirical grounding, leading to confusion between conjectures and substantiated results. Second, the overlap between high-induction and high-FV heads needs clearer analysis. Also, claims about the novelty of studying large-scale models are weakened by prior work. Overall, the work would benefit from more rigorous evidence, tighter framing of conclusions, and more explicit discussion of building upon these insights for practical solutions to further improve ICL.

**Additional Comments On Reviewer Discussion:**

The reviewers' major concerns remain unresolved.

---

### Decision · Program_Chairs · 2025-01-22

Reject